# The utility of earth science information in post-earthquake land-use decision-making: the 2010-2011 Canterbury earthquake sequence in Aotearoa New Zealand

Mark C. Quigley[1,2], Wendy Saunders[3], Chris Massey[3], Russ Van Dissen[3], Pilar Villamor[3], Helen Jack[4], Nicola Litchfield[3]

[1]School of Earth Sciences, The University of Melbourne, Parkville, 3010, Australia

[2]School of Earth and Environment, University of Canterbury, Christchurch, 8140, New Zealand

[3]GNS Science, Lower Hutt, 5040 New Zealand

[4]Environment Canterbury, Christchurch, 8140, New Zealand

*Correspondence to*: Mark C. Quigley (mark.quigley@unimelb.edu.au)

**Abstract.** Earth science information (data, knowledge, advice) can enhance the evidence base for land-use decision-making. The utility of this information depends on factors including the context and objectives of land-use decisions, the timeliness and efficiency with which earth science information is delivered, and the strength, relevance, uncertainties and risks assigned to earth science information relative to other inputs. We investigate land-use decision-making practices in Christchurch, New Zealand and the surrounding region in response to mass movement (e.g., rockfall, cliff collapses) and ground surface fault rupture hazards incurred during the 2010-2011 Canterbury earthquake sequence (CES). Rockfall fatality risk models combining hazard, exposure and vulnerability data were co-produced by earth scientists and decision-makers and formed primary evidence for risk-based land-use decision-making with adaptive capacity. A public consultation and submission process enabled consideration of additional earth science information, primarily via stakeholder requests. For fault rupture hazards, pre-disaster geotechnical guidelines and collaboration networks enhanced the ability of earth scientists to rapidly acquire relevant observational data to meet the demands of decision-makers. Expeditious decision-making granted permissive consent for rebuilding in the fault rupture zone based on preliminary scientific advice that was subsequently supported by more comprehensive geological investigations. Rapidly fluctuating and diverse demands for post-disaster earth science information may be best met through prior establishment of (i) land-use policies and technical guidelines tailored for a variety of diverse disaster scenarios, (ii) hazard and risk analyses in land-use plans, including acquisition of geospatial and other earth science data, and (iii) co-ordinated science networks that may comprise sub-groups with diverse goals, operational perspectives and protocols, which allow the many facets of science information acquisition and delivery to be successfully addressed. Despite the collective knowledge shared here, some recent land use practices in New Zealand continue to prioritize other (e.g., socioeconomic) factors above earth science information, even in areas of extreme disaster risk.

## 1. Introduction

Evidence-based, participative and equitable land-use planning is considered best-practice for balancing risk reduction actions with opportunities for sustainable land use. Land-use planning can reduce the exposure and vulnerability of humans and infrastructure to hazards (Mader et al., 1980; Birkmann, 2006). Ideally, land-use planning is used to reduce risk prior to the occurrence of a hazard event (Johnson et al, 2005; Becker et al, 2010; Schwab, 2014). However, in practice the occurrence of a disaster is often the greatest stimulus for developing and implementing policies for land-use change (Saunders and Becker, 2015). Risk-based land-use planning actions could include temporary- to long-term changes to land-use policies, building codes and regulations, and occupancy conditions, including voluntary or compulsory land acquisitions to reduce or prohibit future occupation. Buildings and infrastructure may be removed, relocated, and / or redeveloped. Land-use planning policies are developed and implemented by local government bodies, with relevant information sought from, and policy developed with, indigenous peoples, critical infrastructure representatives, experts, consultants, stakeholders, advisory committees, and participating citizen community groups (Comerio, 2013; Platt and Drinkwater, 2016).

Relevant land-use decision-making inputs may be sourced from economic, insurance, engineering, architectural, planning, and other socio- and scientific disciplines including the earth sciences. How these inputs are individually assessed and evaluated against each other may not be well documented or transparent in decision-making processes. Scientific information may be variably solicited by and co-produced with decision-makers, to independently produced and contributed via other parties (e.g., stakeholders, media) (Quigley et al., 2019a,b). Divergent information within and across disciplines may be evaluated and prioritized systematically (e.g., expert elicitation; Colson and Cooke, 2018), more qualitatively (e.g., expert caucusing, Wright, 2014), selectively (e.g., sole consultation with or prioritization of information from trusted providers) or intuitively (Klein, 2008). There may be no simple relationship between the "best science", "more information", and "better decisions" (Sarewitz and Pielke, 2001). Analyses of natural variability and randomness inherent in science information (i.e., aleatoric uncertainties) and uncertainties originating from incomplete knowledge of natural phenomena, data, systems, and models (i.e., epistemic uncertainties) (e.g., Mullins et al., 2016) may or may not be fully understood, undertaken by science providers, and effectively communicated to decision-makers (Quigley et al., 2019a). Perspectives on uncertainties and risks, and their applicability to decision-making may vary significantly amongst scientists and decision-makers (van Asselt, 2000). Decision-makers may choose to reduce, acknowledge, and/or ignore uncertainties based on their characteristics and the nature and context of decision-making (Lipshitz and Strauss, 1997). Other (i.e., non-scientific) inputs and considerations such as time, resource, and political pressures may be prioritized (Quigley et al., 2019a). Enacted decisions may not align with prevailing science evidence (Gluckman, 2014) and decision-making processes may be variably complex, non-linear, bureaucratic, and even illogical (Davis, 2014).

Evaluating the specific role(s) of earth science information across the spectrum of diverse land-use planning decisions is challenging (e.g. Saunders and Beban, 2014). Here we focus on describing the components and temporal characteristics of the earth science response to major earthquakes during the 2010-2011 Canterbury earthquake sequence (CES) in New Zealand's South Island. Our objective is to analyze how diverse types of earth science information pertaining to ground surface fault rupture and mass movements in populated areas was (or was not) used in post-disaster land use planning decisions, and why (or why not). We limit the scope of this paper to these hazards, in which we, the authors, were direct participants and possess intimate knowledge, although brief

comparisons are made to CES liquefaction-related land use planning decisions. A thorough analysis of the use of earth science in informing land use decision-making is provided by Quigley et al., (2019a,b) and thus we do not duplicate their efforts here.

In this paper, the term 'mass movements' is used as a generalized, collective descriptor for rockfalls, cliff collapse, boulder rolls, soil and rock-slides, toe slumps, and any other gravitational movements of rock, sediment and / or soil (following Massey et al., 2013). We retain this term whilst noting that the Christchurch District Plan uses the term 'slope instability management area' to define separate hazard domains for each of the dominant hazard types such as cliff collapse, rockfall or boulder roll, and other mass movements (https://ccc.govt.nz/assets/Documents/The-Council/Plans-Strategies-Policies-Bylaws/District-Plan/New-Christchurch-district-plan/CDP-Chapter-05-Aug-2017.pdf). Other globally utilized classification schemes use the broad term 'landslide' to include the mass movements described herein (e.g, Cruden and Varnes, 1996; Hungr et al., 2014).

Because we conduct this analysis from the perspective of scientist participants and observers of this process, rather than from the perspective of a researcher not directly involved, or as decision-makers or personally-affected stakeholders, we acknowledge the potential for cognitive biases (e.g., actor-observer bias, focus principle) including heuristics (e.g., availability heuristic, fundamental attribution error) (e.g., Jones and Nisbett, 1971; Schwarz et al., 1991; Korteling at al., 2018) to influence our individual and collective perceptions of how earth science information did or did not influence decision-making processes. To cope with this potential issue, we deliberately confine this study to presentation and analysis of evidence for:

*(i) earth science information acquisition (what was acquired, who acquired it, when it was acquired),*

*(ii) earth science information communication (who communicated what to whom, how and when was it communicated), and*

*(iii) earth science utility in decision-making (what was and was not provided to decision-makers, what was and was not acknowledged by decision-makers to have been used in decision-making).*

Our overarching goals are three-fold:

1) to identify lessons learned from (i) to (iii);

2) to highlight pressure points in (i) to (iii) (e.g., bottlenecks that slow down processes or otherwise reduce earth science information transfer between scientists and decision-makers) for focusing future research efforts pertaining to science information acquisition, communication, and enhancement of decision-making utility; and

3) to share observations of earthquake hazard-related land use decisions elsewhere in New Zealand and to briefly compare them to CES processes described herein.

We do not attempt to analyze attributes of the decision-makers, their internal processes, the roles of other non-scientific inputs that may have influenced (e.g., biased) decision-making (e.g., Eppler, 2007), and / or the (positive and negative) societal impacts of enacted decisions. For discussions of some of these aspects we turn readers to the Greater Christchurch Group (2017), Johnson and Mamula-Seadon, (2014) and Tyson (2016).

In New Zealand, many of the major regulatory documents and guidelines for addressing aspects of seismic hazard, such as earthquake loading standards (NZS 2004) and dam safety guidelines (NZSOLD 2015), are formulated around probabilistic seismic hazard assessment (PSHA). The prevalent use of PSHA in New Zealand has

influenced the formulation of other earthquake hazard related documents such as the Ministry for the Environment's "Active Fault Guidelines" (Kerr et al., 2003) which uses surface rupture recurrence interval (as a proxy for annual exceedance probability). For landslide hazards there are no New Zealand-specific regulatory documents to guide the practitioner with respect to the appropriate method to analyse the hazard or what risk metrics to use and/or what risk thresholds a regulator should consider as intolerable/tolerable. Within New Zealand, Saunders and Glassey (2007) provided guidelines on landslide hazard and risk for planners (with use applications relevant to geological and geotechnical practitioners, and developers) for resource consent applications and planning documents at regional and district levels. The Australian Geomechanics Society (AGS) 2007 landslide risk management guidelines are widely used within New Zealand but not officially adopted by any agency. The AGS started to develop guidelines for landslide risk management in the 1990's and subsequently updated in 2007 (AGS, 2007). At the same time and in conjunction with the AGS, the Joint Technical Committee-1 (JTC-1) comprising members from the International Society for Soil Mechanics and Geotechnical Engineering (ISSMGE), the International Society for Rock Mechanics (ISRM), and the International Association of Engineering Geologists (IAEG) developed guidelines on landslide susceptibility, hazard and risk zoning for land use planning (Fell et al., 2008a), to set recommended international best practice. To a large extent, this is what has been adopted in the mass movement hazards work covered in this investigation.

**2. The Canterbury Earthquake Sequence**

The CES commenced with the 4 September 2010 moment magnitude (Mw) 7.1 Darfield earthquake (Figs. 1 and 2). The epicenter was located approximately 44 km west of the central business district of Christchurch (Gledhill et al., 2011) (Fig. 1A). Peak ground accelerations (PGAs) of 0.15 to 0.3 g were recorded in central Christchurch (Bradley et al., 2014). A 29.5 ± 0.5 km long ground-surface rupture along the Greendale Fault was produced across the Canterbury Plains (Quigley et al., 2012) (Fig. 1A). Strong ground shaking resulted in widespread and locally severe liquefaction and lateral spreading in eastern Christchurch, Kaiapoi, and in isolated areas throughout the region (Cubrinovski et al. 2010; Quigley et al., 2013; Townsend et al, 2016) (Fig. 1B). Mass movements were isolated to sparsely populated areas and did not cause significant damage to infrastructure (Stahl et al., 2014; Khajavi et al., 2012). Damage to unreinforced masonry structures and some infrastructure (e.g., water and sewerage pipes) occurred (Dizhur et al. 2010; Hughes et al. 2015). Direct damage costs were estimated at $3.34 B USD (Berryman, 2012) but no fatalities occurred.

The aftershock sequence following the Darfield earthquake included the 22 February 2011 Mw 6.2 Christchurch earthquake that caused 185 fatalities. PGAs of 0.3 to 0.6 g were recorded in central Christchurch (Bradley et al., 2014). Severe damage to buildings and infrastructure, and major-to-severe liquefaction and mass movements occurred (Dellow et al., 2011; Kaiser et al., 2012; Massey et al., 2014). Direct damage costs were estimated at $13.34 B USD (Berryman, 2012). The Mw 6.0 June aftershock ($1 B USD), and Mw 5.9 December aftershock ($17.25 M USD) also caused more damage, liquefaction and mass movements, but no fatalities. More than 1200 buildings in central Christchurch were ultimately demolished. The estimated total loss exceeds from the CES is $31 B USD (http://www.nbr.co.nz/article/christchurch-quake-costrises-10b-40b-bd-139278). A detailed summary of the diverse range of geological and environmental impacts of the CES is provided by Quigley et al. (2016).

**3. Governance arrangements**

In response to the earthquake sequence, a complex system of organizations, roles and responsibilities emerged, as visually represented in Fig. 2. This complexity is typical of most recovery efforts from natural disasters, with the greatest challenge being able to manage this complexity effectively, both for those inside and outside the system (Controller and Auditor-General, 2012). It was within this complexity that the science response needed to operate. An understanding of the different functions and responsibilities was required.

Of those agencies presented in Fig. 2, the primary entry points for science was through key agencies summarized in Table 1.

**Table 1: Summary of agency roles and responsibilities at the time of the CES initiation (Controller and Auditor-General, 2012)**

| Agency | Role in Canterbury response and recovery |
|---|---|
| Canterbury Earthquake Recovery Authority (CERA) | <ul><li>leading the recovery, including overall monitoring of the recovery;</li><li>managing the Crown's buying of residential properties in the red zone;</li><li>leading, through the Christchurch Central Development Unit, the rebuilding of Christchurch's CBD;</li><li>co-funding and co-managing the repair and rebuilding of infrastructure (with Christchurch City Council and the New Zealand Transport Agency</li><li>providing policy advice to the Minister about land zone decisions; and</li><li>working with insurers to monitor and encourage the timely settling of insurance claims.</li></ul> |
| Local authorities (Christchurch City Council, Selwyn District Council | <ul><li>repairing horizontal infrastructure;</li><li>repairing amenities that they own and operate;</li><li>planning land use;</li><li>issuing building and planning consents;</li><li>supporting the community by providing information, advocacy, and other help;</li><li>being CERA's Recovery Plan strategic partner; and</li><li>for Christchurch City Council, preparing the initial recovery plan for Christchurch's CBD</li></ul> |
| Regional authority (Environment Canterbury) | <ul><li>lead agency behind the Greater Christchurch Urban Growth Strategy. Parts of the strategy were fast-tracked to bring forward available land for new residential areas to replace those that have been zoned red.</li><li>repairing and maintaining river management and drainage schemes throughout the region for flood protection purposes.</li><li>processing resource consents for earthworks, discharges to land and water, and disposal of waste and rubble to landfills.</li></ul> |
| Earthquake Commission (EQC) | <ul><li>administer the insurance against natural disaster damage provided for under the Earthquake Commission Act (EQC handles residential claims, not commercial claims);</li><li>help research and educate about matters relevant to natural disaster damage;</li><li>manage the Natural Disaster Fund, including arranging reinsurance;</li><li>managing the assessing and processing of residential contents, building, and land claims;</li><li>overseeing the home repair programme, through a contract with The Fletcher Construction Company Limited;</li><li>delivering the winter heating programme; and</li><li>funding Geonet's participation in the Engineering Advisory Group.</li></ul> |

| | |
|---|---|
| Ministry of Business Innovation & Employment (formally Department of Building & Housing) | • policy advice on housing, which included six main areas: how to rebuild in greater Christchurch; housing supply and market response; consenting systems and process – how to build on land in Canterbury; information and monitoring – how to measure rebuilding; the national effects of the Canterbury earthquakes on earthquake-prone buildings; and the Canterbury Earthquakes Royal Commission;<br>• publishing revised building standards;<br>• testing solutions to land remediation and foundation designs;<br>• technical guidance on building in earthquake-prone areas;<br>• providing a sector education and training programme to help a quality rebuild;<br>• providing the Canterbury Earthquake Temporary Accommodation Service in partnership with Ministry of Social Development; and<br>• reporting on the structural performance of buildings in Christchurch |
| Department of Internal Affairs | • Through the Ministry of Civil Defence and Emergency Management (MCDEM), the Department of Internal Affairs had overall responsibility for the Civil Defence and Emergency Management Act 2002;<br>• lead agency for reducing risk and building community readiness nationally, and for emergencies;<br>• overseeing reviews of the civil defence response to the earthquakes;<br>• administering the Canterbury Earthquake Appeal Trust; and<br>• providing policy advice. |
| Department of the Prime Minister & Cabinet (DPMC) | • lead role in co-ordinating the national emergency response in support of the Ministry of Civil Defence & Emergency Management;<br>• DPMC's Policy Advisory Group and the Cabinet Office supported the setting up of a new Cabinet committee; and<br>• supported CERA, mainly through providing staff to CERA. |

**4. Timeline of science response and decision-making**

The earth science response to the Darfield earthquake (~4:35 a.m. NZST) commenced almost immediately, with the rapid organization and deployment of local university- and government-based earth scientists and engineers to the field by NZST 7 a.m. (Fig. 3; Table 2). Earth science information was communicated to the media within hours of the earthquake (Table 2). The government research institute GNS Science (https://www.gns.cri.nz/) responded under the GeoNet programme (https://www.geonet.org.nz/). Scientists and geotechnical and structural engineers from across New Zealand responding to this earthquake were collectively coordinated by the New Zealand Natural Hazards Research Platform (Beaven et al., 2017). The Natural Hazards Research Platform actions were aimed at documenting earthquake effects and associated land and infrastructure damage, and communicating information to local authority officials, officials from the national insurer against natural hazards (NZ Earthquake Commission, EQC), private insurers, emergency managers, and central government agencies (Berryman, 2012). The earth science response expanded and diversified with continued seismic activity and associated impacts throughout the CES (Berryman, 2012).

In Fig. 3, key actions by scientists and decision-makers for (i) fault rupture response and (ii) mass movement response are plotted against time and seismicity (magnitude ≥ 3.0 earthquakes). Selected significant government actions pertaining to these hazards are also shown. We describe the details of fault rupture (Section 4) and mass movement (Section 5) below and in Tables 2 and S1, respectively, and describe and interpret key observations from the timeline in Section 6.

**Table 2. Fault rupture response timeline: list of scientist and decision-maker actions**

| Date (NZST) | Event | Event |
|---|---|---|

| | Type | |
|---|---|---|
| 4/09/2010 | **SEIS** | Darfield earthquake |
| 4/09/2010 | SA | University of Canterbury (UC) rupture mapping team begin co-ordination |
| 4/09/2010 | SCA | Earth scientist undertakes radio interview on earthquake impacts |
| 4/09/2010 | SA | UC rupture mapping team deployed |
| 4/09/2010 | SA | GNS Science (GNS) rupture team deployed |
| 4/09/2010 | SCA | Earth scientist interviewed on TVNZ on earthquake impacts |
| 4/09/2010 | SA | GNS/Geonet response team and UC rupture team leaders co-ordinate |
| 4/09/2010 | SA | Ground surface rupture located and initial mapping commences |
| 5/09/2010 | SA | Formation of UC-GNS rupture mapping team and mapping initiated |
| 5/09/2010 | SA | GNS team inspects aerial photographs for pre-Darfield earthquake evidence |
| 6/09/2010 | SCA | GNS press release - Canterbury Fault Had Not Ruptured For At Least 16,000 Years |
| 10/09/2010 | SCA | UC-GNS rupture team leaders present maps to meeting of Federated Farmers |
| 11/09/2010 | SFA | Lidar acquisition (flight date) |
| 18/09/2010 | SFA | Field mapping finishes |
| 20/09/2010 | SCA | Scientists receive lidar data |
| 1/11/2010 | DMRA | Selwyn District Council (SDC) begins to seek advice from consultant on rebuilding in fault zone |
| 17/11/2010 | DMRA | GNS is asked by farmer about rebuild, GNS contacts Environment Canterbury (ECan) with proposal to produce a report |
| 2/12/2010 | SA | Consultant supplies SDC with preliminary estimate of fault recurrence interval class, no fault avoidance zones mapped |
| 7/12/2010 | SCA | First international peer-reviewed publication of Greendale Fault map |
| 18/01/2011 | DMRA | ECan and SDC seek advice from GNS on fault recurrence interval class and fault avoidance zone mapping |
| 21/01/2011 | SCA | GNS provide letter of expert advice on fault recurrence interval class to SDC |
| 25/01/2011 | SFA | Ministry of Civil Defence & Emergency Management reimburses ECan for post-earthquake lidar as a response cost |
| 16/02/2011 | DMA | SDC issues first building consent for new domestic dwelling in fault zone |
| 22/02/2011 | **SEIS** | Mw 6.2 Christchurch Earthquake |
| 4/03/2011 | DMA | SDC issues building consent - New Domestic Dwelling in Fault Zone |
| 10/03/2011 | DMRA | SDC requests information from GNS re. location of temporary building site relative to Greendale Fault for earthquake-affected Christchurch residents |
| 11/03/2011 | DMA | SDC issues building consent - Relocated Domestic Dwelling |
| 17/03/2011 | SCA | GNS provide letter of expert advice to SDC on proposed location of temporary housing near Greendale Fault |
| 30/03/2011 | DMA | SDC issues building consent - Relocated Domestic Dwelling |
| 19/05/2011 | SCA | GNS / ECan Report Published: Greendale Fault: Investigation of Surface Rupture Characteristics for Fault Avoidance Zonation |
| 15/06/2011 | DMA | SDC issues building consent - Demolition Of Domestic Dwelling And new domestic dwelling |
| 1/07/2011 | DMA | SDC issues building consent - Replacement Garage |
| 11/07/2011 | DMA | SDC issues building consent - Dwelling Repairs |
| 13/09/2011 | DMA | SDC issues building consent - Domestic Dwelling & Garage |

| 13/10/2011 | DMA | SDC issues building consent - Demoltion Of Dwelling & Relocated Dwelling |
|---|---|---|
| 6/12/2011 | DMA | SDC issues building consent - Relocated Dwelling & Carport |
| 31/05/2012 | DMA | SDC issues building consent - Domestic Dwelling |
| 8/08/2012 | DMA | SDC issues building consent - Demolition Of Domestic Dwelling & New Domestic Dwelling |
| 15/08/2012 | DMA | SDC issues building consent - Domestic Dwelling |
| 3/09/2012 | SCA | Publication of Greendale Fault avoidance zone map (Villamor et al., 2012, NZJGG) |
| 5/09/2012 | SA | Paleoseismic trenching of Greendale Fault commences (site 1) |
| 3/10/2012 | DMA | SDC issues building consent - Domestic Dwelling Additions & Domestic Garage |
| 23/10/2012 | DMA | SDC issues building consent - Domestic Dwelling |
| 9/11/2012 | DMA | SDC issues building consent - Domestic Dwelling |
| 21/11/2012 | SCA | Media Article Published In Press "Dig Shows Another Quake Was On Fault" |
| 5/03/2013 | SA | Paleoseismic trenching of Greendale Fault site 2 |
| 1/06/2014 | SCA | GNS Report: Paleoseismology of the 2010 Mw 7.1 Darfield Earthquake Source, Greendale Fault |
| 16/10/2014 | SCA | Publication of Hornblow et al (2014) Paleoseismology of the 2010 Mw 7.1 Darfield earthquake source, Greendale Fault |
| 18/05/2015 | SCA | ECan updates SDC on revised recurrence interval class for Greendale Fault |
| | | |

**Event Type**: **SEIS:** MAJOR SEISMIC EVENT; **SA:** SCIENCE ACTION; **SCA:** SCIENCE COMMUNICATION ACTION; **DMRA:** DECISION-MAKER REQUEST FOR SCIENCE ADVICE; **DMA**: DECISION- MAKER ACTION; **SFA:** SCIENCE FUNDING ACTION

190

## 5. Ground surface fault rupture

*5.1. Earth science response*

Co-ordinated field mapping of the Greendale Fault ground surface rupture by University of Canterbury and GNS Science earth scientists commenced on 5 September 2010 (Fig. 3, Table 2). Rupture mapping included data

195  collection via real-time-kinematic GPS instruments and compass-and-tape measurement techniques, entered nightly into GIS, and supplemented with aerial surveys by helicopter. Preliminary field maps of the surface rupture trace were made publicly available on GNS Science and individual websites (Quigley and Forte, 2017), and presented to affected parties (i.e., property owners in the fault zone and surrounding area) within six days of the Darfield earthquake. Six residential dwellings were damaged by ground surface rupture on the Greendale Fault

200  (Van Dissen et al., 2011) in the 2010 Darfield earthquake. A power substation was impacted by the ground surface fault rupture but was repaired and is still in use. Four agricultural structures (implement or dairy sheds) were impacted by surface fault rupture but none were demolished.

An independent inspection of historical aerial photographs to identify whether any surface evidence for pre-2010 (predecessor) ground surface ruptures on the Greendale Fault was evident was undertaken immediately by GNS

205  Science. A GNS Science press release published on 6 September (GNS Media Release, 2010) stated that the "*Canterbury fault had not ruptured for at least 16,000 years*" based on an absence of evidence for pre-2010 surface faulting and assumptions that the land surface was post-last glacial in age (Forsyth et al., 2008). These comments featured in national and international newspapers on 7 September 2010 (Fig. 3).

A proposal to the Environment Canterbury Regional Council (hereafter referred to as Environment Canterbury) by the Natural Hazards Research Platform to fund the acquisition of airborne LiDAR data over the Greendale Fault for the purposes of fault mapping was submitted within days of the Darfield earthquake. LiDAR data was collected on 11 September 2010, as part of a larger scale LiDAR acquisition program over urban Christchurch, with a primary focus on observing land surface elevation changes in liquefaction-affected areas.

A collaborative University of Canterbury and GNS Science rupture mapping team was under significant time-pressure to map the fault rupture traces because many landowners had commenced land repairs that removed surface evidence for faulting. By the time the processed LiDAR data was available to the team (20 September 2010) the field mapping program had been completed and much of the evidence of surface rupture had been removed or modified. The LiDAR data was useful for validating field measurements (Litchfield et al., 2014), obtaining better constraints on distributed deformation, and producing final fault surface rupture maps (Villamor et al., 2011, 2012). Fortuitously, pre-earthquake LiDAR data (obtained for the purposes of regional flood mapping) was also available for small isolated sections of Greendale Fault, thereby enabling LiDAR differencing to be used to characterise high-resolution ground rupture displacements for one of the first times globally (Duffy et al., 2013).

The first peer-reviewed articles to present fault rupture maps were published in December 2010 (Quigley et al., 2010a,b) but these were insufficiently detailed for developing fault avoidance zone maps that would be consistent with available planning guidelines (Kerr et al., 2003). Public talks, reports to government agencies, media appearances, and research publications provided a diverse and effective communication platform that reached stakeholders and decision-makers.

By November 2010, the Selwyn District Council ("SDC" in Fig. 3) recognized the need to obtain expert advice on the location and approximate recurrence intervals of surface rupture on the Greendale Fault, to assist them and owners of earthquake-damaged properties to better understand the spatial and temporal context of this hazard when considering rebuilding strategies. In New Zealand, it is a territorial authority's (city or district council's) responsibility under the Resource Management Act 1991 to set land use policies and rules in their district plan for managing development on or near active faults (Kerr et al, 2003). The Selwyn District Council initially commissioned an independent consultant to provide this advice; general advice on fault zone width and preliminary estimates of recurrence interval was given (Fig. 3) but fault avoidance maps were not provided. Environment Canterbury ("ECan" in Fig. 3) commonly contributes technical information, planning and management advice, and funding to district councils for issues pertaining to geological hazards. Stimulated by increasing desire from property owners to gain certainty over rebuilding criteria, Environment Canterbury began to discuss the production of fault avoidance maps and likely recurrence interval class of the Greendale Fault with GNS Science (Fig. 3). GNS Science provided Environment Canterbury with a preliminary letter of recurrence interval class and fault avoidance mapping of the eastern end of the fault rupture (Villamor and Litchfield, 2011) that was urgently requested by decision-makers to inform siting of temporary housing for citizens displaced from Christchurch city after the 22 February 2011 earthquake). Subsequently, Environment Canterbury commissioned GNS Science to produce a detailed map of the fault avoidance zone (Fig. 3), in accordance with best-practice guidelines (see next section) (Kerr et al., 2003). Fault avoidance zone maps (Fig. 1C) were provided to the Selwyn District Council and Environment Canterbury from GNS Science. Building consent for the first domestic building proximal to the fault zone was approved prior to receipt of this report. A series of consents for demolition,

relocation, new construction, repairs, and amendments to dwellings were issued by the Selwyn District Council
beginning in March 2011 (Fig. 3).

*5.2 Land use decision-making*

In 2003 the New Zealand Ministry for the Environment (https://www.mfe.govt.nz/) and their research partners developed the Active Fault Guidelines (Kerr et al., 2003) to assist planners, emergency managers, earth scientists, and people in the building industry to reduce exposure to, and mitigate against, ground-surface fault rupture
hazards. The Active Fault Guidelines provide a risk-based approach to combine ground rupture hazard parameters with a hierarchical classification of different types of engineered structures ('Building Importance Category') for developed sites and those for which development is planned. Surface rupture hazard of an active fault is defined by two parameters: 1) the location and complexity of surface rupture of the fault (Well defined, Distributed, or Uncertain; Fig. 1C), and 2) the activity of the fault, as measured by its average recurrence interval of surface
rupture (i.e. the Recurrence Interval Class). Villamor et al. (2011) proposed a conservative estimate of recurrence interval of ground surface rupturing earthquakes on the Greendale Fault of >5000 to ≤10,000 years (Recurrence Interval Class IV). The Active Fault guidelines recommend residential development be permitted in fault zones with this recurrence interval class.

Based on the increasing strength of evidence, beginning with the initial statement from GNS Science on 6[th]
September 2010, and supported by consultant advice provided in November 2010 and the Villamor et al. (2011) report, Selwyn District Council decided to permit building in the fault avoidance zone but to make property owners aware of the location of the Greendale Fault on their properties. The fault appears in the current Selwyn District Plan for information purposes only, and the following note is added to all Project Information Memorandum (PIM) applications (see https://www.building.govt.nz/building-officials/guides-for-building-officials/project-
information-memoranda/ for definition of PIM) and/or asked for as a part of the Building Consent process if there is no PIM application:

*"The proposed building is located on a site where the Greendale Fault and associated fault deformation may be present. Most building and development work will still be able to be undertaken despite the possible presence of the fault, however a report from a geotechnical engineer will be required to confirm the suitability of the building*
*site, with this report informing the foundation design."*

This note is added to all properties identified within the fault avoidance zone represented on the fault map (see examples, in Figures 1C and D). All buildings within the fault avoidance zone would require *"specific engineering input to verify that the structural design of the building has considered the proximity of the fault line"*. The fault avoidance zone was established based on information contained the Active Fault guidelines, and contains the
ground deformation zone and the 20m setback. Of the six houses damaged by ground-surface fault rupture, two were demolished and rebuilt on the same site (in the fault zone); one was demolished and rebuilt within 40 m of the original site (outside of the fault zone); one was demolished and not rebuilt; and two houses were repaired on site (in the fault zone). Temporary housing for hosting displaced citizens from Christchurch was built outside the fault avoidance zone.

Paleoseismic trenching investigations of the Greendale Fault were funded by a contestable research grant written by scientists and funded by the government insurer EQC, rather than the agency responsible for land use decision-making (Selwyn District Council). Paleoseismic investigations began in 2012 and provided evidence for a surface rupturing earthquake on the Greendale Fault approximately 21,000 to 28,000 years ago (Hornblow et al., 2014a).

This was interpreted by Hornblow et al. (2014a) to represent the timing of the penultimate earthquake. The fault was re-assigned to a more permissive recurrence interval class (Class V: 10,000 to 20,000 years) by Van Dissen et al. (2015) and Hornblow et al. (2014b) but this revision had no implications for previously-enacted property consent decisions. The paleoseismic information was consistent with previously-provided earth science information and validated the Selwyn District Council decision-making, but was not explicitly used in decision-making around rebuilding damaged houses because *(i)* it was not available at the time of decision-making, and *(ii)* given the tectonic setting and preliminary evidence-of-absence for events in the last >5000 yr, the need to expedite this research and / or delay decision-making was not deemed necessary by decision-makers nor advocated for by scientific experts.

The Selwyn District Council's desire to make scientifically justifiable and expedient decisions pertaining to (re)building in the Greendale Fault zone was enabled by (i) the pre-event existence of guidelines for building on or near active faults; (ii) the rapid mapping of the fault ground surface rupture and fault avoidance zones; and (iii) the rapid estimation of fault recurrence interval class. Selwyn District Council was not aware of, and / or was not implementing fault avoidance guidelines in to planning practice prior to the Darfield earthquake, and thus this change is attributed to experiences associated with the Darfield earthquake. Land use decision-making was undertaken with partial knowledge that increased with time (i.e., the actual recurrence interval had not yet been determined from paleoseismic investigations and statistical analyses of the fault) and was informed by prevailing expert advice and new scientific data (e.g., fault maps). The permissive land use classification placed final decision-making control in the hands of the individuals, who could choose to rebuild in the fault zone or not, depending on their personal perceptions of risk and other factors (e.g., existing infrastructure, insurance considerations). It is important to note that even if the Greendale Fault location and timing of the last earthquake had been known prior to the Darfield earthquake, the Ministry for the Environment guidelines (Kerr et al, 2003) would have supported a permissive approach to land use activities, and fault-rupture associated losses may not have been reduced. This represents the spatiotemporal coincidence of very low probability event with a risk-based approach to land use planning. Selwyn District Council are currently reviewing their District Plan (as of 16 March 2020) in the context of Ministry for the Environment guidelines and are proposing restrictions on new Building Importance Category 4 and 5 buildings within the fault avoidance zone. This shows how post-disaster information may be used to inform adjustments to land-use decision frameworks even a decade or more after the catalyzing event, subject to availability of purpose-specific earth science information and planning structure(s) (guidelines, ordinances) for that information to be utilized.

*5.3 Summary and implications*

Pre-disaster planning guidelines (Kerr et al., 2003) and collaboration networks enhanced the ability of earth scientists to rapidly acquire relevant field data, to identify and seek funding for the acquisition of other potentially relevant data (LiDAR), and to ultimately meet the expedient requirements of decision-makers. Permissive consent for rebuilding in the surface fault rupture zone favoured expeditious decision-making based on preliminary scientific advice, rather than delaying decisions until uncertainties could be reduced through more comprehensive geological investigations (e.g, paleoseismic investigations). Paleoseismic data would have been more relevant in decision-making processes had it been acquired prior to the Greendale Fault rupture, been acquired immediately following the earthquake, and/or was considered by decision-makers to be an essential component in decision-

making. However, the absence of clear ground-surface evidence for preceding events (there was no pre-existing scarp), the post-Darfield earthquake focus on other activities (fault mapping, responses to ongoing seismicity), and the expert advice that long recurrence intervals were expected for this fault, provide explanations for why paleoseismic investigations did not occur before, or immediately after this earthquake.

Maps of past surface rupture characteristics and paleoseismic data exist for many faults in New Zealand (e.g., Langridge et al., 2016), including the most hazardous faults with high probability future events (e.g., Alpine Fault - Langridge et al., 2018; Berryman et al., 2012; Wellington Fault - Rhoades et al., 2011), and some of the faults that ruptured in the 2016 Mw 7.8 Kaikoura earthquake (Little et al., 2018). How this information informs contemporary land use practises is highly variable and important to consider. Successful implementation of fault

avoidance zones to reduce exposure to anticipated surface rupture hazards on the active faults in the Wellington area have been enacted by various councils in the region (e.g. Perrin and Wood, 2003; Van Dissen and Heron, 2003; Van Dissen et al., 2005, 2006a, 2006b).  Many district councils around New Zealand (n ≥ 11) have fault avoidance zones in their district plans, or are including active faults when reviewing their plans with associated fault avoidance zone provisions. In contrast, despite well characterized scientific information that indicates ~20

to 50% 50-year conditional probabilities of a ground-rupturing earthquake on the Alpine Fault (Biasi et al., 2015) and advice that its mapped trajectory through populated regions define a major risk to life, dwellings and infrastructure (Orchiston et al., 2018; Zorn et al., 2018), in 2016 local decision makers (Westland District Council) voted against enforcing a fault avoidance zone due to public concerns including perceived inadequate compensation for relocations, property devaluations, and autonomy over personal choice

(https://www.stuff.co.nz/national/87604696/restrictions-on-building-along-alpine-fault-in-franz-josef-scrapped). Richardson (2018) suggests this represents an example of how challenging it may be to enact anticipatory (i.e., pre-disaster) deliberative policy making on issues that may be subject to inter-generational equity problems.

        In the instance of the 2016 Kaikoura earthquake, most of the fault ruptures occurred in sparsely populated or
mountainous areas (Litchfield et al., 2018).  Although many of the faults had been previously mapped at a regional scale, detailed fault avoidance zones had not been mapped due to the high cost and low risk posed by these faults. In areas where damage to houses from strong shaking, fault rupture and/or landslides required rebuilds and repairs, fault avoidance zones have been mapped for any nearby fault traces (whether or not they ruptured during the Kaikoura earthquake) to help inform the rebuilding process.  These fault avoidance zones are currently being

incorporated into the Hurunui and Kaikoura district plans.  Until this planning process is complete, which can take several years, there is nothing to stop people from rebuilding on a fault trace, however in most cases people are rebuilding away from the trace because there is enough available land to do so in these rural areas, and because people do not want to experience damage to their house again.

While many councils in New Zealand were and still are in the process of mapping faults and fault avoidance zones and incorporating these into district plans through their 10-yearly review process, the CES and Kaikoura earthquakes have reinforced, particularly to elected representatives and affected communities, the usefulness of building away from active fault traces. The opportunities and challenges associated with disaster risk reduction through fault zone avoidance are complex and depend on how earth science interacts with a variety of

socioeconomic considerations and planning processes, including how fault-rupture hazards and risks compare with other types of risks (e.g., flooding) in the hazardscape. The Active Fault Guidelines, while now 17 years old,

did provide a useful framework for risk-based planning around the active faults. However, a review and possible updating of the guidelines is now considered advantageous, based on changes in legislation, improved understanding of active faults, and technological advances. This would also ensure the new generation of planners implement lessons from the CES, many of which may not be aware of the extent of consequences and planning implications that arose.

## 6. Mass movement response

### 6.1. Earth science response

The GNS Science-led landslide team (*https://www.geonet.org.nz/landslide/how*) deployed to Canterbury in response to the Darfield earthquake on 7 September 2010 as part of the GeoNet science response. Initial news reports, residential accounts, and preliminary field observations indicated mass movements triggered by this earthquake were rare and localised, with minimal impact on people and infrastructure. The two largest mass movements were in the Harper Hills ("HH" in Fig. 1A), where coseismic ground cracks with up to 1 m displacement were identified (Stahl et al., 2014); and at Castle Rock, where a debris avalanche of about 100 $m^3$ travelled down slope and reached the northern entrance of the Lyttleton road tunnel (Fig. 1F). Minor rockfalls and boulder displacements of small (< 1 m diameter) rocks with site-specific volumes of 5 to 60 $m^3$ occurred in isolated areas of the Port Hills (Massey et al., 2017; Khajavi et al., 2012).

The 22 February 2011 Mw 6.2 Christchurch earthquake generated PGAs proximal to rockfall sites of up to 2.2 g (Kaiser et al., 2012; Bradley et al., 2014; Massey et al., 2015). Widespread mass movements occurred, including disrupted landslides such as rockfalls, rock, soil and debris slides and avalanches and cliff-top cracking in soil and rock and toe-slope cracking in soil, all associated with coherent landslides such as slides and slumps (see Keefer 2002; Hungr et al., 2014 for definitions of these terms) (Dellow et al., 2011; Massey et al., 2014, 2017; Carey et al., 2017). Rockfalls and debris avalanches (locally referred to as boulder rolls and cliff collapses, respectively) constituted the most abundant and hazardous mass movements and posed the highest risks to people and buildings. Over 6,000 mapped rockfalls were triggered by the 22 February 2011 earthquake (Fig. 1F). Rockfalls and debris avalanches severely impacted 100 residential dwellings (Taig et al., 2015). Rapid assessments of future risk underpinned the immediate evacuation of 560 dwellings, which later reduced to 456 in July 2011 as a result of reassessing the risk from the hazards present at each dwelling (Macfarlane and Yetton, 2013). Three fatalities occurred in residential areas (one person within their home and two on properties adjacent to collapsing cliff faces) due to cliff collapse. Two fatalities occurred due to impacts from falling rocks in non-residential areas of the Port Hills (Massey et al., 2014).

The GeoNet landslide response to the Christchurch Mw 6.2 earthquake commenced with immediate planning operations. On 23 February, two teams of geologists and engineering geologists were deployed to Christchurch to provide information that could be used by Civil Defence and Emergency Management personnel to evaluate risks to, and protect, life safety. The teams immediately began working with the Urban Search and Rescue (USAR) team; the partnership was rapidly initiated because a member of the landslide response team was in the initial USAR team deployed to Christchurch. Initially, the teams concentrated on field mapping rockfall boulders and trails, ground cracking, cliff collapse volumes, and debris inundation areas. Mapping was initially done using pre-Darfield earthquake aerial imagery and LIDAR ground models but evolved over time to include post-Christchurch

earthquake aerial and terrestrial LIDAR data, aerial and satellite imagery and InSAR data. In addition to the field mapping, slope instrumentation was also rapidly deployed to monitor ground shaking, permanent ground movement, rainfall, pore-water pressures and sub-surface movement on several slopes across the Port Hills, where ongoing slope movement posed a risk to people and lifelines. The landslide response team worked with the USAR team in the Port Hills to help identify landslide hazards in those areas where people had been killed; to assess safe access for emergency responders and to monitor any developing landslide hazards that could pose a risk to people and lifeline infrastructure post the 22 February earthquake. A third team carried out an aerial survey of the main area affected by the earthquake to help identify areas affected by landslides and to obtain an assessment of the "scale" of the landslides affecting the hill suburbs (Hancox and Perrin, 2011; Hancox et al., 2011). This information was fed back to the response teams on the ground.

Within days of the 22 February 2011 earthquake, the GeoNet landslide response teams began working together with geotechnical engineers and engineering geologists from local engineering consultancies, and with University of Canterbury staff and students. To coordinate the response to the geotechnical issues caused by the earthquake in the Port Hills, this group of technical people, including the GeoNet landslide response teams, formed the Port Hills Geotechnical Group (PHGG). The PHGG went on to be contracted by Christchurch City Council. The Christchurch City Council quickly recognized the benefits to be gained from accessing scientific information from the PHGG and they organized daily meetings, chaired by a local engineering geologist, who coordinated the geotechnical response and fed back information to the Emergency Operations Centre (Macfarlane and Yetton, 2013). The PHGG organized the geotechnical response teams to respond to requests for help and/or advice coming into the Emergency Operations Centre. These requests were "triaged" via the PHGG meetings and tasks were given to the various teams. The PHGG fed back advice and information to the Emergency Operations Centre, which was used to help the appropriate authorities to evacuate homes (initially via the placement of "red placards" and later via the placement of Section 124 notices under the 2004 Building Act, http://www.legislation.govt.nz/act/public/2004/0072/latest/DLM307300.html) where the risk from mass movement hazards (referred to in the district plan as 'mass movement', 'cliff collapse', and 'rockfall' management areas) was assessed in the field as being too high. The decision to evacuate people from such homes was validated in the 13 June Mw 6.0, 23 December 2011 Mw 5.9, and 14 February 2016 Mw 5.7 earthquakes in Christchurch, when many of these houses were further impacted and damaged by cliff collapses and/or rockfalls (Massey et al., 2016). Rockfall trajectories also traversed formerly occupied sites, where houses had been demolished. In addition to the GeoNet landslide response teams on the ground, GNS Science also provided Geographical Information System (GIS) support to the Christchurch City Council and the PHGG, which included collating geospatial information being fed back via the response teams, to define the locations and types of the various hazards being identified (Macfarlane and Yetton, 2013). After the initial response phase, these tasks passed to the Christchurch City Council. The processes followed by the GeoNet landslide response teams and the PHGG are discussed in detail by Yates (2014).

PHGG-led responses continued for about one year after the 22 February 2011 earthquake (Fig. 2), with increases in activity after the 13 June and 23 December 2011 aftershocks. During this period, the PHGG divided the Port Hills suburbs into sectors and appointed a lead consultant for each sector (Macfarlane and Yetton, 2013). GNS Science was assigned an advisory role for all sectors and teams. The critical data collected by the teams in the field comprised the locations and volume of rockfalls and debris that fell from the slopes. In addition, mapping of the cracking associated with permanent displacement of the ground referred to in the Port Hills as "toe slumping"

(a type of coherent landslides, described by Keefer, 2002) was also carried out. All mapping data was made available via the Christchurch City Council's GIS server, and then later remotely via a remote mapping application

so that field teams could more easily add data whilst in the field. It was during this period that the Christchurch City Council initiated area-wide systematic mass movement assessments initially focusing on rockfall and cliff collapse hazards, and later including ground cracking. These assessments were led by GNS Science, working with the PHGG, and they relied upon data collected by the PHGG. The PHGG consultants also field verified the results from the area-wide assessments and were involved with all aspects of the assessments.

In July 2012, paleoseismic investigations of prehistoric rockfall boulders commenced at Rapaki (Fig. 1F; Fig. 3). The first major research paper on pre-historic (middle Holocene) rockfalls of comparable severity and extent to the CES rockfalls was published in 2014 (Mackey and Quigley, 2014), more than 2 years after initial mass-movement related land use planning decisions (see Fig. 3, and next section). A major research program continued at this site until 2019 (e.g., Borella et al., 2019). A relevant finding for rockfall hazard was that CES rockfalls

traveled further than analogous pre-historic predecessors because intervening anthropogenic deforestation on the hillslopes reduced boulder-tree impacts; the conclusion was that native reforestation in the Port Hills, once well established, could potentially reduce the spatial dimensions of rockfall hazard (Borella et al., 2016). Investigations at other sites highlighted possible spatial variations in the timing of rockfall events (Litchfield et al., 2016) and importance of other geological and seismological variables (Borella et al., 2019) that cautioned against applying

the findings from Rapaki more broadly without site-specific investigations.

*6.2. Land use decision-making*

The GNS Science landslide teams working with the PHGG on the area-wide assessments of landslide hazard and risk were contracted by both the Christchurch City Council for the area-wide assessments and the Canterbury

Earthquake Recovery Authority (CERA) for specific tasks relating to the area-wide assessments. These agencies had different and some similar tasks that they were responsible for. Decision-making processes that relied upon the area-wide assessments of mass movement hazard and risk in the Port Hills are described separately below.

*6.2.1 Christchurch City Council*


Following the 22 February 2011 earthquake, the immediate focus of the Christchurch City Council was on identifying the risk to life and lifelines from earthquake hazards. Dwellings in the Port Hills were assessed by the PHGG based on field-based qualitative analysis of the hazard and risk that the hazards may pose to their occupants. PHGG personnel would then recommend to Christchurch City Council placement of "red placards"

for areas of high perceived risks. If a red placard was placed on a dwelling, it meant that the risk was assessed as being too high for the occupants to continue to live there. Many of these placards that were placed on dwellings for mass movement hazards eventually became Section 124 Notices (Building Act, 2004). These notices were reviewed many times by different technical people within the PHGG and a few (about 8 of the 400 placed) were challenged by residents via a Ministry of Building, Innovation and Employment (MBIE) hearing process.

Running parallel to this work was an area-wide mass movement hazard and risk assessment. The initial phase was to establish what the Christchurch City Council wanted to use as the risk metrics. Christchurch City Council chose

the annual individual fatality risk (AIFR) and GNS Science adopted the Australian Geomechanics Society Landslide Risk Assessment Guidelines (AGS, 2007) as the methodology to estimate this level of risk and its uncertainty (Massey et al., 2014). The area-wide assessments identified some areas in the Port Hills where the risk from individual mass movements affected smaller localised communities. Site-specific risk assessments were carried out for these areas, again adopting the AIFR as the risk metric of choice. This work generated over 20 reports (https://ccc.govt.nz/environment/land/slope-stability/), which were all independently peer reviewed by both the GNS Science appointed independent review panel, and the Christchurch City Council appointed independent peer reviewer.

These reports and their results in map form were ultimately used by the Christchurch City Council to define their High Hazard Areas within their replacement District Plan. The Council endeavoured to "*make Christchurch more resilient to shocks and stresses*" (https://greaterchristchurch.org.nz/projects/resilient-greater-christchurch/) and Christchurch joined the "Resilient Cities Network" (https://www.100resilientcities.org/cities/) in December 2013. An agenda setting workshop was held in March 2014 with broad participation from central and local government agencies, the private sector, academic institutions and community organisations. Given the significance of various natural hazards identified as shocks and stresses in the workshop, the Natural Hazards Strategy and the District Plan provisions were seen by Council as critical components of their broader resilient city framework (Beaumont, 2015).

To derive the High Hazard Areas, Christchurch City Council adopted various input parameters for the risk models. They established, using individual and community meetings with affected Port Hills stakeholders to discuss hazard and risk modelling results, risk thresholds to define the different High Hazard Areas and the planning rules associated with them. This process included a hearing process where High Hazard Areas defined by the Christchurch City Council were adopted in their replacement District Plan as cliff collapse management areas 1 and 2, rockfall management areas 1 and 2, mass movement areas 1, 2 and 3, and a category for the remainder of the Port Hills and Banks Peninsula slope instability management area.

The independent hearing process gave people an opportunity to raise their concerns on the Christchurch City Council's proposed replacement District Plan. Some submitters brought in experts and additional data that they considered relevant to their submissions (see Quigley et al., 2019a). The key issues raised in the 13 submissions relating to the mass movement risk assessments carried out by GNS Science for Council were (Massey, 2015a; 2015b):

1. the appropriateness of carrying out area-wide risk assessment for mass movement hazards;
2. the appropriateness of the parameters adopted in the risk assessments; and
3. the uncertainties associated with the risk estimates and the perceived "conservatism" associated with the adopted parameters used in the risk assessments.

The term "conservative" in the context of natural hazard risk could be interpreted as meaning "safe" or "too safe" where the opposite would be "unsafe or too unsafe". The level of "conservatism" in the risk estimates depends on the choice of values associated with each parameter used in the different risk models. The uncertainties can drive the risk estimates in both directions, "higher and lower" and their impact on the risk results were quantified in the GNS Science reports CR2011/311 (Massey et al, 2011) and CR2012/214 (Massey et al, 2012). The uncertainties did not have equal weighting in the risk analyses. The uncertainties given in the GNS Science reports span the

range of values associated with each parameter in the risk analyses. Therefore, if a rockfall risk model were to adopt all of the lower "optimistic" values, and the results compared to a risk model adopting the upper "pessimistic" values – from the ranges considered by GNS Science to be reasonable – there would be slightly more than one order of magnitude (a mean factor of 30, where a factor of 10 is an order of magnitude) difference between the results of the rockfall risk models presented in the GNS Science reports. For planning purposes it is not appropriate to use risk models that adopt all of the lower "optimistic" values as this would lead to "unsafe" or "too unsafe" decision making.

The findings of the hearing commissioners are contained in Hansen et al. (2015). Prior to the hearing, expert caucusing was carried out to try to find agreement between the various experts representing the submitters. This generated a document (Experts Joint Statement, detailed in Hansen et al., 2015). An important statement made in this document, which was signed by all experts, stated:

> *We acknowledge the possibility that future earthquakes have the potential to cause additional rockfall and cliff collapse in the Port Hills. Published, peer-reviewed geologic data do not exclude the possibility of future rockfall triggering events from the ongoing sequence or other seismic events. Available site-specific geologic data suggest that clusters of severe rockfall events may be separated by hiatuses spanning 1000s of years but further analysis from additional sites is required to test this hypothesis. The seismicity model was developed by an international expert panel using international best practice and has undergone peer review. Given the recent and modelled earthquake clustering activity and the large uncertainties on predicted ground-motion for an individual earthquake, we agree that the level of conservatism is appropriate.*

Hansen et al. (2015) decided that the area-wide risk assessments were appropriate, and they supported the risk-based approach to natural hazards management in the proposed replacement District Plan. Hansen et al. (2015) noted the risk management approach and the level of conservatism was also accepted by those present at the expert caucusing. A provision, initially raised by GNS Science and later endorsed by all experts, was that there should be a way in which local site-specific information for a dwelling could be used to re-evaluate the landslide risk, thus allowing the given site to be reclassified (e.g., changing its High Hazard Area status through a certification process). The independent hearings panel agreed to this provision provided that any re-assessment should follow the same method and approach adopted for the area-wide assessments.

### 6.2.2 Canterbury Earthquake Recovery Authority (CERA)

Concurrently with the area-wide mass movement hazard and risk assessments, CERA developed a land zoning policy. Central government identified the need to assist people in the worst-affected suburbs who were otherwise facing protracted individual negotiations with their insurers. In the Port Hills, rockfall fatality risk modelling by GNS Science, commissioned by the Christchurch City Council, was the primary geotechnical resource that was used to inform land zoning decisions (Jacka, 2015). Further numerical modelling and property specific geological and geotechnical information from PHGG and other engineering consultants was also used by CERA to inform their decision making, although the details of this information utility are not known to the authors of this paper. This process eventuated with the categorisation of dwellings into either a "green" zone or "residential red zone", within which the Government would offer to buy properties from the owners. The term "residential red zone" was used to distinguish it from the red zone in the Christchurch central business district ("CBD red zone") that was cordoned with restricted access controlled by the NZ Defence Force during the state of emergency

immediately after the 22 February 2011 earthquake; and from "red placarding" of buildings deemed to be dangerous or insanitary and thus posing a risk to human safety (i.e., issued a Section 124 notice) (Jacka, 2015).

The area-wide process for categorising properties into green zones or residential red zones was not intended by CERA to be a formal RMA hazard -zoning or hazard-mapping tool, in either the Port Hills or on the flat land.

CERA reviewed various area-wide risk mitigation approaches, such as hazard avoidance through removal of dwellings, to the design of engineering mitigation works such as debris retention structures to prevent rocks and landslide debris from hitting homes. It was generally found that in most cases, effective engineering solutions were considered infeasible for assorted socioeconomic reasons; some of the initial designs for such structures appeared to "fence in" entire communities and were thus not considered desirable from a social structure and

aesthetic perspective, and the total costs of maintenance and replacement of retention structures throughout their life span (e.g., in response to impacts by geological debris) were significant (Richards, 2012).

Upon consideration of these aspects, the New Zealand Cabinet (a collegiate body of senior ministers including the Prime Minister that operate with collective responsibility) enacted a policy decision to use the area-wide hazard and risk assessments carried out by GNS Science as the main tools to identify dwellings that would be

eligible for the residential red zone offer. The policy decision made by Cabinet adopted an annual individual life risk threshold of $10^{-4}$ (1 in 10,000 chance of being killed per year) as the maximum acceptable level of life risk from landslide hazards in the Port Hills. To identity such properties, CERA adopted slightly different input parameters for the risk models to those adopted by the Christchurch City Council. The life risk models developed by GNS Science include assumptions that vary the level of "conservatism" in the AIFR calculations. These

include: the percentage of time an individual is in a dwelling; that seismic activity is likely to decay with time (acknowledging that earthquake frequency and magnitude are expected to decrease with time after the 22 February 2011 earthquake); and whether or not residents are evacuated following a major aftershock, and therefore not present and exposed to landslides from subsequent earthquakes and rain events. The different input parameters used in the risk models and chosen by CERA and Christchurch City Council are shown in Table S3. These

differences could be attributed to the Christchurch City Council decision to take a more precautionary approach relative to CERA to ensure they did not increase exposure in areas of known hazard over long (>5-10 yr) "planning" timescales, whilst CERA wanted to give people an option to move on more quickly post-earthquake and thus did not hold future planning-related issues as the predominant input.

On 1 October 2012, the New Zealand Cabinet confirmed the following criteria to be used for residential zoning decisions in the Port Hills (taken from Jacka, 2015):

1. Green zone: where the AIFR $<10^{-4}$, and where land damage and any life risk $\geq10^{-4}$ could be addressed on an individual basis
2. Red zone: where

a. AIFR $\geq 10^{-4}$ adopting the model assumptions in Table S3; or

     b. There is potential for immediate cliff collapse or landslide caused or accentuated by the Canterbury Earthquake Sequence with associated risk to life; and

     c. An engineering solution to mitigate the life risk is judged not desirable, and it would (amongst over factors): be uncertain in terms of detailed design; and/or be disruptive for landowners;

610       and/or not be timely; and/or not cost effective; and put the health and wellbeing of residents at risk.

CERA established their own peer review panel, who were independent of the GNS Science and the Christchurch City Council peer reviewers, to review the risk models, their underpinning data, and how the results appeared on 615 the ground. This policy development process, along with the Christchurch City Council policy process was documented in the evidence given by the various experts to the Independent Hearings Panel, during the Christchurch replacement District Plan Hearings (http://www.chchplan.ihp.govt.nz/wp-content/uploads/2015/03/310_495-CCC-and-Crown-Joint-memorandum-regarding-Natural-Hazards-hearing-08.12.14.pdf).


*6.3 Summary and implications*

Decision-making processes in the mass movement-affected areas of the Port Hills in Christchurch were aided by diverse teams of earth scientists, engineers, government agents, and other individuals. They were developed under conditions involving large risks and uncertainties, and under intense scrutiny and time-pressure. These aspects 625 placed immense pressure on the technical and policy development teams working to deliver the underpinning science and inform policy developments for the decision makers. It is the opinion of the authors of this paper that the residential zoning process carried out by CERA would have been more efficient and less time intensive if existing policy framework had been in place, which defined the criteria, metrics and methods that could be used for zoning, linked to anticipated government outcomes. Whilst acknowledging that each natural disaster brings its 630 own issues, it is hoped that future disasters will draw on the policy processes that are outlined in this paper. Furthermore, the Christchurch City Council District Plan could have already included landslide hazard zones for the Port Hills, as past evidence of such hazards, especially rainfall-induced landslides and some evidence of earthquake-induced landslide existed prior to the CES. This highlights the usefulness of pre-disaster data and planning to provide a context for future hazards and risks so that the severity and impact of future events are not 635 'unexpected' from a government point of view.

We posit that provisions and tools under the Resource Management Act 1991 (currently under review) should include what risk metrics are to be used to underpin hazard zoning, and what processes should be followed to develop community-based risk tolerability thresholds for life, building, economic, cultural, and wider societal risks. Debate about what a 'tolerable' individual risk threshold actually is, including whether it should be micro-640 zoned to capture spatial variations in societal risk, are likely to be more managed outside of disaster response periods, when a variety of other pressing concerns may influence risk analysis and judgements.

A further recommendation is that MBIE, with technical societies, should develop guidelines for practitioners carrying out risk analyses, to ensure the methods adopted are transparent and follow international best practice. Enhanced coherence in the treatment of risk between the Resource Management Act 1991, the Building Act 2004, 645 the Civil Defence Emergency Management Act 2002 and the Earthquake Commission Act 1993 could enable post-disaster decision-making to be more expedient, transparent, and defensible. Many of these aspects transcend application to mass movement hazards to potentially apply to natural disasters in their broadest sense.

**7. Discussion**

*7.1 Interpretation of science response timeline*

The demand for earth science information connected with the CES, as proxied by the frequency and diversity of scientist actions (data acquisition, information communication, organizational development and co-ordination), is non-linear and highly concentrated into short (ca. 1 month) time-windows immediately following major earthquakes. These time-windows commonly correspond with enhanced seismicity rates and (in two instances) government-actioned state of emergency declarations. Prior studies have established that major disasters significantly compress the time available for policy and other decision making (Johnson and Mamula-Seddon 2014; Olshansky et al. 2012; Fordham 2007; Beavan et al., 2017). Throughout the CES, scientists had to work rapidly to generate and communicate earth science information that was directly relevant for decision-making, whilst attempting to produce peer-reviewed scientific outputs to achieve scientific credibility and legitimacy (Sarkki et al. 2014; Parker and Crona 2012; Hackett 1997; Fordham 2007). GNS Science is a crown-owned company required to conduct scientific research for New Zealand's benefit [Sections 4 and 5.1(a), CRI Act 1992], including a mandated need to respond to natural disasters as well as contracted obligations under the GeoNet programme. In this instance, the acquisition of fundamental (and often transient) observational data was imperative for both emergency response and land-use planning considerations. To ensure effective transfer of scientific information to decision-makers and stakeholders, communications including meetings, emails, verbal communications, and delivery of technical reports were deliberately prioritized above rapid authorship of international peer-reviewed science articles. We consider that maintenance of diverse collaborative networks, comprising industry and university-hosted researchers, increased the reach and breadth of communication activities that were able to be undertaken in this time-compressed environment (Quigley and Forte, 2017; Quigley et al., 2019a,b). We consider that some of the time-pressure related challenges of post-disaster science response may be addressed through establishment of strong pre-disaster relationships amongst diverse sub-groups with distinct goals, operational perspectives and protocols.

On a related note, Fig. 3 also reveals how decision-makers and stakeholders sought earth science information from diverse perspectives and for diverse reasons; for example, from independent consultants (e.g., "*SDC seeks consultant advice on rebuilding in FZ*") and/or from scientists operating in a basic, rather than for applied (science for policy) science perspective (e.g., "*Stakeholders request evidence from independent studies / experts*"). Earth science information was variably sought prior to and during major decisions, and both prior to and after independent publication of research. In some cases, publication of research in peer-reviewed scientific journals (e.g., Massey et al., 2014; Hornblow et al., 2014a) post-dated related decision-making (e.g., building consents, land zoning) by > 1-2 years. Many mass movement research papers from the CES were co-authored by researchers from government research institutes, universities, and industry (e.g., Massey et al., 2014). The ongoing maintenance of diverse research teams is proposed to be beneficial from the perspective of awareness of this research (although, see Section 4.3). The conclusions here are restricted primarily to the acquisition and communication of earth science information. As thoroughly described in Beaven et al. (2017), a lack of balance between research and policy input in any science boundary organization may limit higher-level science integration with policy.

*7.2 Other evidence for earth science utility*

In addition to the evidence presented above, two other types of evidence indicate scientific information was used and valued in CES land-use planning decision-making: documented acknowledgements from decision-makers, and the alignment of enacted decisions with prevailing science evidence and advice.

With reference to science provisions to central government-led land use policy development and decision-making during the second reading of Canterbury Earthquake Recovery Bill in April 2011 (http://www.legislation.govt.nz/bill/government/2011/0286/latest/versions.aspx) the Minister for CERA Hon. Brownlee stated "…*The decisions that need to be made here are very, very dependent upon research about the condition of the land in Christchurch, and upon getting enough information to deal with individuals who have those broken properties so that they can be given some choices about what their future is ...*" (https://www.courtsofnz.govt.nz/cases/quake-outcasts-and-fowler-v-minister-for-canterbury-earthquake-recovery/%40%40images/fileDecision). In 2012, he told the Christchurch Press that "…*I'd love to be able to fix all of that [earthquake land issues] for people immediately, [but] we've got to get the science and engineering right on how to progress…*" (http://www.stuff.co.nz/the-press/news/7656654/Brownlee-fed-up-with-moaning-residents). In 2013, he told the Christchurch Press that "*We know from the extensive ground-truthing and area-wide modelling that the risk of rock roll in this part of the Port Hills is high; hence the need to zone the land red…*" (http://www.stuff.co.nz/the-press/news/8220906/I-told-you-so-says-Brownlee-on-rockfall). After another earthquake in the region in 2016, the Minister said, "*The decay curve provided a timeline for how long it would take for a particular sequence of earthquakes that we went through to settle down. The decay curve said right out to the 30 year horizon, you can expect a declining amount of seismic activity. Periodically there may be a seismic shake that's...a little larger than others*" (https://www.stuff.co.nz/national/politics/76869793/earthquake-minister-gerry-brownlee-says-shake-was-expected).

With reference to the Replacement Christchurch District Plan, it was formally acknowledged by the Christchurch City Council and central government that the proposed plan *"is based on complex technical modelling and outputs*" that rely on "*geotechnical and scientific background research*" and that the "*most effective approach*" for "*refining the issues*" that could arise from submitters wishing to challenge decisions within the plan was "*for relevant experts to enter into technical caucusing on the modelling approach and methodology*" prior to "*evidence exchange*" in hearings (http://www.chchplan.ihp.govt.nz/wp-content/uploads/2015/03/310_495-CCC-and-Crown-Joint-memorandum-regarding-Natural-Hazards-hearing-08.12.14.pdf).

Independent science evidence on rockfall hazard (Quigley et al., 2019a,b) that was submitted to the Replacement Christchurch District Plan panel by stakeholders was deemed to be "*of assistance to the Panel*", who "*urge[d] [those researchers] to work continue to further the current level of understanding*" to "*support…a regime that would allow hazard lines to be adjusted when better information becomes available…*" (http://www.chchplan.ihp.govt.nz/wp-content/uploads/2015/03/Natural-Hazards-Part.pdf).

Environment Canterbury commissioned investigations on the Greendale Fault for fault avoidance zonation (ECan report R11/25; Villamor et al 2011), provided the report to the Selwyn District Council in July 2011, and provided science updates and links to emerging science content for several years after the earthquake (e.g., May 2015) (Fig. 2). The last Environment Canterbury letter to the Selwyn District Council states, "*It is my understanding that SDC has been providing people wishing to build on or close to the Greendale Fault with information on the location of the fault avoidance zone, and that people have been voluntarily building away from the fault trace, regardless of the very low likelihood of movement on the fault in the near future. This approach is reasonable, and we recommend that land owners and potential land owners continue to be informed of the location of the fault*

*avoidance zone in order to make their own decision about purchasing or building"* (Jack, 2015). Selwyn District Council provides science information including maps of the fault avoidance zone to land users (see above 'Decision-making' section pertaining to fault rupture) and is now proposing to incorporate some fault avoidance provisions for critical facilities within the Greendale fault avoidance zone into its District Plan as part of its 10-yearly plan review process. Clearly, the decision-making process has been directly informed by, and aligns with,

prevailing scientific evidence and advice.

     There are many divergent viewpoints about aspects of the government response to the Canterbury earthquakes, particularly in terms of interactions between local and national government agents, of politicization of aspects of the earthquake recovery ([https://essay.utwente.nl/70392/1/Neth_BA_BMS.pdf](https://essay.utwente.nl/70392/1/Neth_BA_BMS.pdf)), of the balance between top-down vs. community-driven recovery and policy development activities (Comerio, 2013); and of the lack of

transparency in how specific science and engineering inputs contributed to land use planning in liquefaction-affected areas (Quigley et al., 2019a,b). Nonetheless, from the perspective of authors of this paper (specifically limited to the surface rupture and mass movement studies presented herein), decisions pertaining to land use policy development, property-by-property decisions, and decision appeal processes was well informed by earth science inputs and ultimately aligned with prevailing science evidence.

*7.3 Comparison with earth science utility in liquefaction-related land use decision-making*

     In contrast to the clearly evidenced and prominent role of earth science inputs in informing mass movement and fault ground surface rupture decision making, the role of earth science in the CERA-led land-use planning decisions for liquefaction affected properties is less certain.

     As reported in Quigley et al. (2019a), CERA's decision-making statement for liquefaction properties states: *"If*

*the estimated cost of reinstating the land to its pre-earthquake condition, up to a maximum value capped by the estimated value of the land ("EQC contribution"), plus the estimated cost of raising the land to an elevation such as to consent with the CCC [Christchurch City Council] building code ("betterment cost–raising of land"), plus the estimated cost of mitigating against lateral-spreading effects that could occur in future earthquakes ("betterment cost–perimeter treatment"), plus the estimated cost of removing and replacing damaged*

*infrastructure (e.g. roads, sewerage, potable water, power infrastructure), exceeded the value of the land (the 2007 capital value of entire property minus improvements), then the area was red zoned. 'Red-zone boundary maps' were constructed by engineering experts but were effectively contour maps based on economic inputs".*

     Given the recurrence of severe liquefaction events during the CES (e.g., Quigley et al., 2013) and associated damage to land and dwellings, the government decision to 'red-zone' many residential areas was logical, albeit

appearing to be most strongly motivated by expediency and economics rather than acquisition and consideration of potentially relevant science inputs. This conclusion is similarly evidenced by the CERA statement "*the urgent need to provide a reasonable degree of certainty to residents in these areas in order to support the recovery process. Speeding up the process of decision-making is crucial for recovery and in order to give confidence to residents, businesses, insurers and investors. This is particularly the case in the worst affected suburbs, where the*

*most severe damage has repeatedly occurred*." (Hon. Brownlee, in Quigley et al., 2019a).

     That said, Hon. Brownlee also stated that *"strength-depth profiles under some parts of Christchurch indicate typically up to 10 metres of 'liquefiable' material. Although some ground settlement may occur, the large reservoir of liquefiable material and these examples suggest that similar characteristics of ground shaking are likely to result in similar amounts of liquefaction in the future"*

(https://ceraarchive.dpmc.govt.nz/sites/default/files/Documents/memorandum-for-cabinet-land-damage-june-2011_0.pdf).

Collectively, this suggests earth science and engineering information including expert-led assessments may have assisted governmental decision-makers in recognizing the need for a land-use policy in the first place, even if the specific roles of these inputs in decision-making is not explicitly evidenced. The variations in approach highlight

the complexities in ensuring a ubiquitously prominent role for earth science in post-disaster decision-making. Even the same decision-makers (e.g, CERA) may prioritize other non-science inputs depending on the nature of the decision and associated relevance of other inputs.

Other differences pertain to decision-makers' perceptions of future risk, consequences of actions leading to an adverse outcome, and associated prioritization of certainty and expediency in delivering decisions to affected

parties. In the case of fault surface rupture, decision-making was informed by expert judgement that assessed the risk of future recurrence as low, the population affected by this hazard was low, and the economic and life safety risks associated with permissive land use consents were low. No deaths occurred due to the Greendale fault surface rupture, even in properties with extreme exposure to this hazard (Fig. 1C) and thus expedient decision-making could occur with minimal risk. In the case of mass movements, the prevailing view was that future hazards could

occur in the short-term, that severe legal, political, economic, and life safety consequences could result from actions leading to permissive land use decisions that exposed humans and infrastructure to future mass movements, and that evidence-backed 'certainty' in decision-making should thus be prioritized above expediency. This, and the desire for a precautionary approach (given CES rockfall-related fatalities and potential for future fatalities), was further highlighted in the CRDP process. The liquefaction-related approach was characterized by

an elevated risk of recurrence in the short term, risks that engineering approaches may not be cost-effective or successfully mitigate against future land damage to desired levels, societal (and possibly political) risks of delaying decisions that affected such a large population (>150,000 properties), and adverse economic, social, logistical, and possibly life-safety consequences arising from permissive land use.

### *7.4 Utility hierarchy in earth science inputs*

The earth science response to the CES included the first New Zealand-based use of:

(i) Light Detecting and Ranging (LiDAR) data to map a historical surface rupture and earthquake-induced ground deformation (Quigley et al., 2010, 2012; Villamor et al., 2011, 2012);

(ii) pre- and post-earthquake LiDAR differencing (Hughes et al., 2015; Duffy et al., 2014) to rapidly characterize earthquake-induced land displacements;

(iii) ground surface fault rupture guidelines (Kerr et al., 2003) in post-earthquake land use planning (Villamor et al., 2011, 2012);

(iv) area-wide 2D and 3D rockfall runout models in New Zealand (Massey et al., 2014; Vick, 2015); and

(v) a risk-based approach (AGS, 2007) to estimate annual individual fatality risks from mass movement hazards (Massey et al., 2014).

Observational data on mass movement and surface rupture hazards and impacts were the most used earth science input in land use planning decisions. The most ubiquitous earth science contributions to land use decision-making during the CES were maps of the spatial distribution and severity of earthquake-induced features (fault surface rupture and mass movements) derived from geospatial data and field observations. Models of future seismicity

that incorporated prior seismic hazard models were imperative for conducting probabilistic evaluations of mass movements (including evaluating possibilities of large, deep-seated landslides) that informed land zoning decision-making by highlighting relatively high risks of future occurrence. The need for expedient decision-making during the CES reduced the utility of paleoseismic research, aimed at characterizing spatiotemporal patterns of CES-type predecessors, because acquisition of paleoseismic data (including laboratory dating of rocks and sediments) required substantive time. Results of these studies were not available rapidly enough to meet the timeline of decision-making, and the available science evidence was perceived to be strong enough to justify decision-making without the need for this information. In the case of fault surface rupture, subsequent paleoseismic data affirmed the permissive approach taken. In the case of mass movements, the enactment of precautionary decision-making with adaptive capacity provides scope for refinement of decisions if additional evidence, including that from paleoseismology (e.g., Mackey and Quigley, 2014; Litchfield et al., 2016) adds sufficiently pertinent information to reconsider land use policies as the short-term seismicity models begin to merge with longer-term earthquake recurrence models. Paleoseismic data obtained elsewhere prior to the CES contributed to seismic risk characterization in greater Christchurch by (i) contributing to the 2010 national seismic hazard model (Stirling et al. 2012), (ii) contributing to seismic hazard models that contributed to short-term seismicity models for Christchurch (Gerstenberger et al., 2104), (iii) informing aspects of the Independent Hearings Panel processes, and (iv) independently-validated recurrence of different seismic shaking intensities. Probabilistic models of future hazard occurrence and risks were used in some instances. Pre-disaster hazard maps (either legally mandated (i.e. hazards zones included in a district plan with provisions) or not (i.e. hazard maps not included in district plan), and geospatial data were infrequently and indirectly used in post-disaster decision-making. Geological (i.e. paleoseismic) investigations were not directly used.

## 8. Conclusions and lessons learned

The authors of this manuscript undertook many different earth science acquisition, communication and planning roles throughout the earth science response to the 2010-2011 Canterbury earthquake sequence. In our opinion, the greatest lessons learnt from our collective experiences, drawn from our diverse perspectives, are:

1. Diverse scientific and operational perspectives and roles, host institutions and career stages (e.g., government-based senior and junior scientists, university-based academics and students), and expertise domains are considered beneficial in enabling earth scientists to collaboratively respond to the diverse and time-compressed demands of this protracted natural disaster (Quigley and Forte, 2017). These aspects could be generally described as promoting a diverse, equitable and inclusive disaster science response culture. Enhancing the roles for indigenous researchers and cultural knowledge in disaster response would further improve this aspect (e.g., Phibbs and Kenney, 2015).

2. Pre-event recovery planning (e.g. having policy in place prior to a disaster that defines the criteria, metrics and methods that could be used for zoning, whilst acknowledging that each natural disaster brings its own issues) would have reduced the considerable time and resource demands on responding earth scientists throughout the CES.

3. In New Zealand, city/district and regional governments currently implement risk-based planning for all natural hazards they may face. In the case of the CES, the District Plan at the time did not include landslide hazard zones for the Port Hills, or for liquefaction, even though evidence of such hazards

existed. Ensuring district plans include hazard zones and risk-based plans is considered beneficial in a global sense. Earth scientists share a responsibility of communicating these hazards and offering inputs relative to risk analyses to appropriate government officials through a variety of actions and approaches (Quigley et al. 2019a,b) that enable design and implementation of these plans.

4. The current New Zealand national planning legislation (i.e. RMA) allows for national policy statements,
environmental standards, or guidance to be developed. In the future, planning for hazards would benefit from one of these tools, which clearly states what risk metrics should be used to underpin hazard zoning, and what processes should be followed to develop community-based risk tolerability thresholds, across a range of risks (e.g. life, building, economic, cultural, social).

5. Technical societies are encouraged to develop guidelines for practitioners carrying out risk assessments,
to ensure the methods adopted are transparent, follow international best practice, and can be readily reviewed and updated. These currently exist in some regions and countries, and not in others.

6. Legislation for land use planning (i.e. RMA), building codes and standards (i.e. Building Act 2004), local government decision making and priorities (i.e. Local Government Act 2002), the national natural hazard insurer (i.e., the Earthquake Commission), and National Disaster Resilience Strategy could be better
aligned, and where practical, share common goals, such as reducing disaster risk, to ensure a consistent approach to risk management.

7. In future events, provision of avenues for rapid funding, to expedite acquisition of high utility geospatial datasets (e.g., LiDAR) would enable a more efficient and rapid science response. These avenues are available in some countries (e.g., the U.S. National Science Foundation's Rapid Response Research
(RAPID) program, which processes applications within 1-2 weeks; Altevogt et al., 2015) but not currently in an official, ongoing, natural disaster-related framework for New Zealand (and other countries).

8. The need for expedient decision-making during the CES negated the utility of paleoseismic research, aimed at characterizing spatiotemporal patterns of CES-type predecessors, because (i) the results of these
time-intensive studies were not available rapidly enough to meet the timeline of decision-making, and (ii) the available science evidence was perceived to be strong enough to justify decision-making in the absence of this data. The greatest potential for the utility of scientific info is if it is available at the time the decisions need to be made. In a post-event "crisis", time-frames are condensed; therefore the best time for scientific information, including paleoseismic data, to be available for disaster risk reduction
decision-making is before forthcoming events, rather than after.

9. Central and local government agencies used earth science inputs to inform decision-making and enact red-zone decisions that clearly reduced further loss (e.g., due to recurrent rockfall and liquefaction in red-zoned areas). Although some CES-related literature has focused on community dissent and the 'slow' recovery in Christchurch, in reality the need for patience (so that relevant science and engineering
evidence can be obtained and considered), pragmatism (decision-making options needed to be of sufficient resolution and simplicity as to be generally applicable to the majority of the populous), and multi-agency co-ordination (in order to holistically resource and utilize science advice in a manner that a more community-led recovery approach may not have been able to do) are additional important factors to consider when analysing the recovery of Christchurch throughout a protracted earthquake sequence.

10. The CES clearly brought into focus that the management of many natural hazards and risks fall largely on the shoulders of those tasked with the responsibility of administering the RMA and issuing resource

and land use consents. In New Zealand, this responsibility is held by regional and district councils. For councils to be able to enact policy aimed at reducing the impacts of ground deformation hazards that successfully stands up to scrutiny in the Environment Court, the basis for decision-making must be

supported by sound evidence from earth science research. If the lessons learned in Christchurch pertaining to fault surface rupture, slope instability, and liquefaction are to be applied elsewhere in New Zealand, then the policies developed by councils to achieve that need to be based on earth science of a similar rigor, breadth, and defensibility as was acquired and utilized throughout the Canterbury earthquake sequence. Beaven et al. (2017) offer numerous insights from a boundary organizational

perspective, including better balancing of policy and research sectors, that would benefit better integration between science and decision-making practice.

*Author contribution*

M.Q. conceptualized this research in consultation with all authors. M.Q. led the writing of the manuscript and created Figures 1 and 3. All authors made substantive content and intellectual contributions to text, figures, and revisions. WS created Table 1 and Figure 2 and made substantive contributions to the governance and land use planning content. M.Q., P.V., H.J., and R.V.D. authored Table 2. C.M., and N.L. authored Tables S1 and S2. M.Q. obtained funding for publication costs.

*Acknowledgements*

M.Q. thanks Port Hills champion Sue Stubenvoll for partial funding for this research and stimulating his involvement in Christchurch Replacement District Plan processes. The authors thank Barbara Dillon and Steven Burnham from Selwyn District Council for provision of information on internal processes pertaining to land use planning consents. Biljana Lukovic provided editable access to reports from which Fig. 1F was derived. The authors acknowledge funding from the MBIE funded Endeavor Programme: Earthquake-induced landscape dynamics.

In writing this paper, our thoughts are with the people of Canterbury who were affected by the earthquakes during 2010-11. It was a traumatic event for many, with substantial loss of life, property, and livelihoods. Many people have experienced ongoing difficulties during the long recovery process. We acknowledge past, present, and future Cantabrians, and hope our science reflections contribute positively to their present and emerging communities.

*Competing interests*

The authors declare that they have no conflict of interest.

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

**FIGURES**

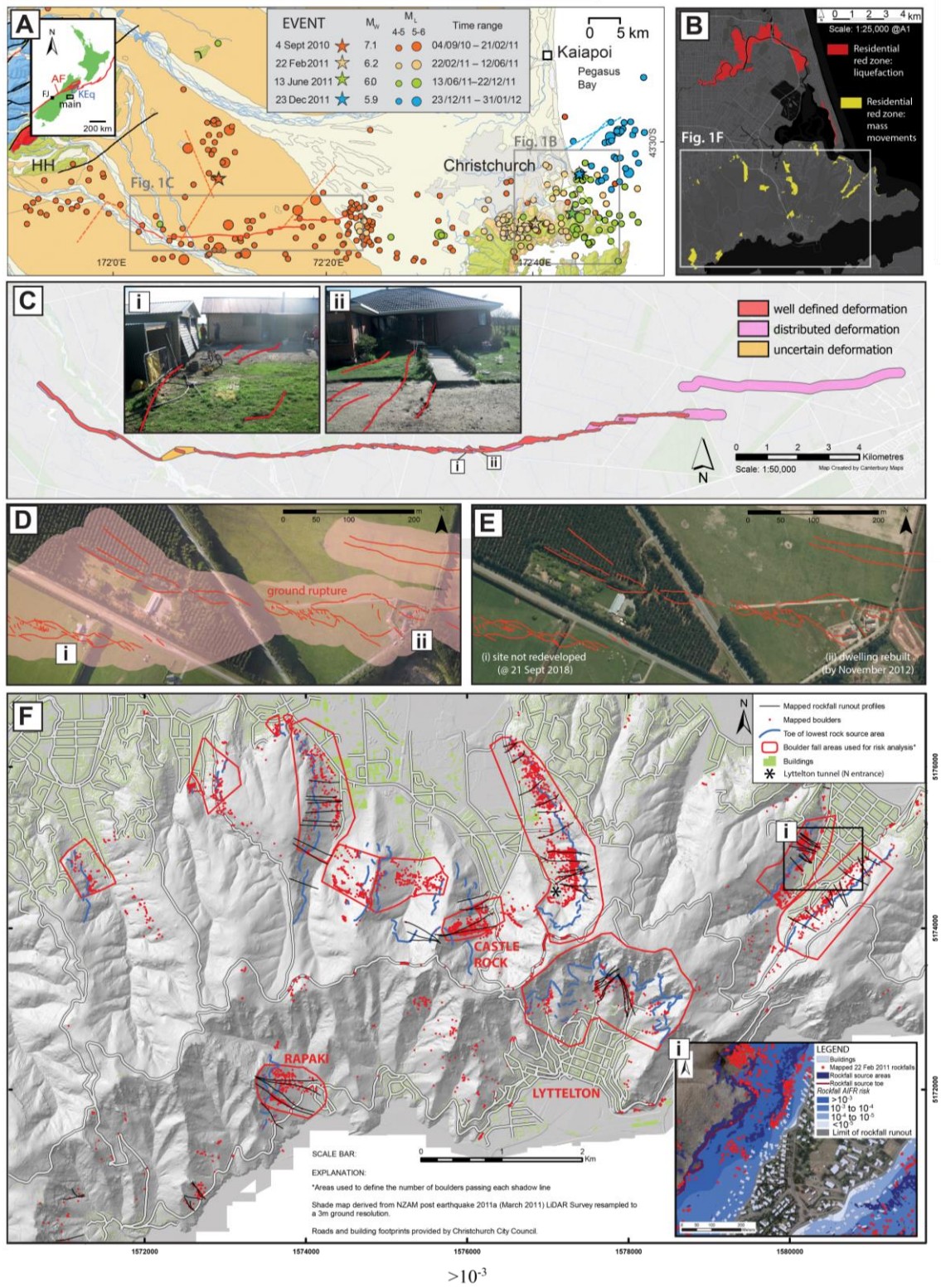

**Figure 1:** (A) Seismicity of the Christchurch, Canterbury region, showing epicentral locations of Mw ≥4.0 earthquakes from 4 September 2010 NZST to 31 January 2012. Locations of the four largest earthquakes shown with stars. Solid red line represents Greendale Fault rupture trace; dashed red lines indicate approximate positions of other source faults of the Darfield earthquake. Other dashed lines (tan, green, blue) indicate approximate positions of source faults for the 22 Feb, 13 June, and 23 Dec. earthquakes, respectively. HH = Harper Hills. Inset image shows location of study area in New Zealand, relative to the Pacific-Australia plate

boundary (red line), the region of the 2016 Mw 7.8 Kaikoura earthquake (KEq), the Alpine Fault (AF), and Franz Josef (FJ). (B) Location of red-zoned residential properties in Christchurch. Red colour denotes properties red-zoned based on liquefaction criteria, yellow colour denotes properties red-zoned based on mass movement (rockfall and cliff collapse) criteria. (C) Fault avoidance zone map for the Greendale Fault, from Villamor et al. (2012), with fault deformation definitions following Kerr et al. (2003). (i) - (ii) ground surface fault rupture traces (red) passing beneath residential structures. (D) Aerial satellite image (imagery date: 15 Feb 2011) with mapped ground surface rupture traces (red lines) and enveloping fault avoidance zones (pink area) for the area enveloping sites (i) and (ii) from (C). (E) Aerial satellite image (imagery date: 7 Nov 2011) showing sites where structures were removed and not rebuilt (site i) and where rebuilding occurred (site ii). Based on imagery inspection, site (ii) remained undeveloped as of 21 Sept 2018. (F) Mapped mass movement hazards (rockfall boulders and source areas, boulder runout trajectories, and areas used for rockfall risk analyses) shown relative to infrastructure (roads - white lines, buildings - green shade) in the Port Hills area of southern Christchurch. Base image is LiDAR DEM hillshade with 3m resolution. Inset (i): Example of AIFR contour map used for decision-making. Figure and inset modified from Massey et al. (2012).

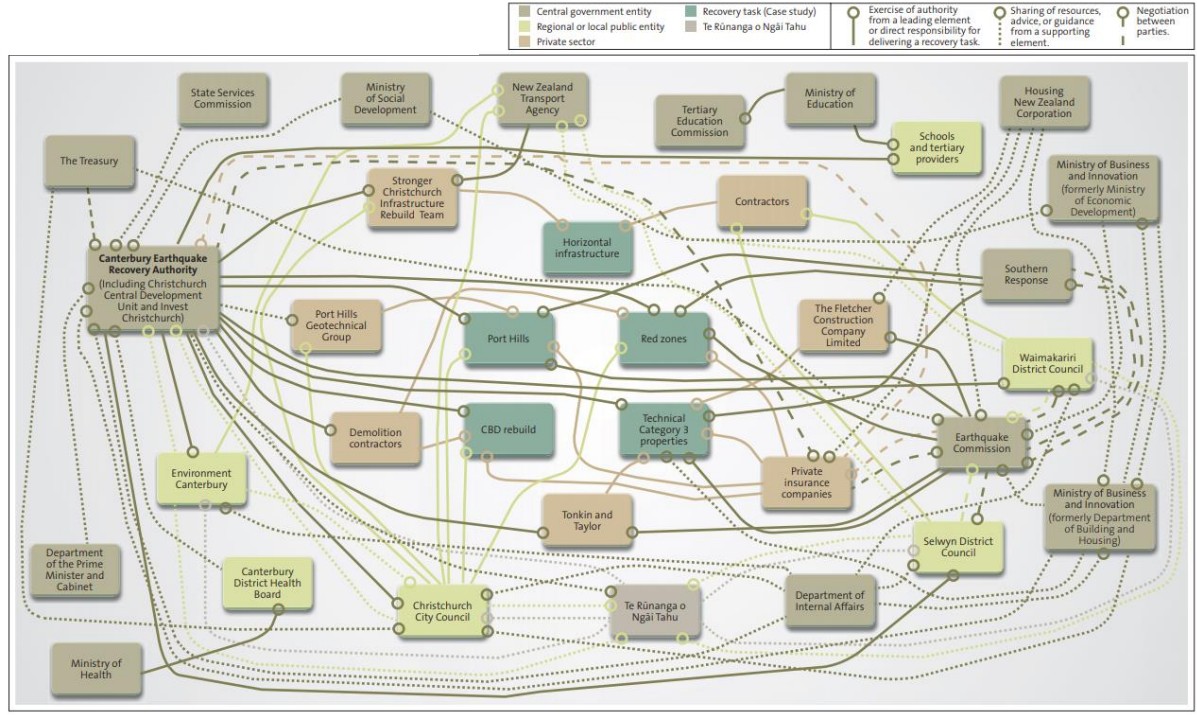

**Figure 2:** Relationships between public sector entities, private companies, Ngāi Tahu, and Canterbury recovery tasks (Controller and Auditor-General, 2012, p16)

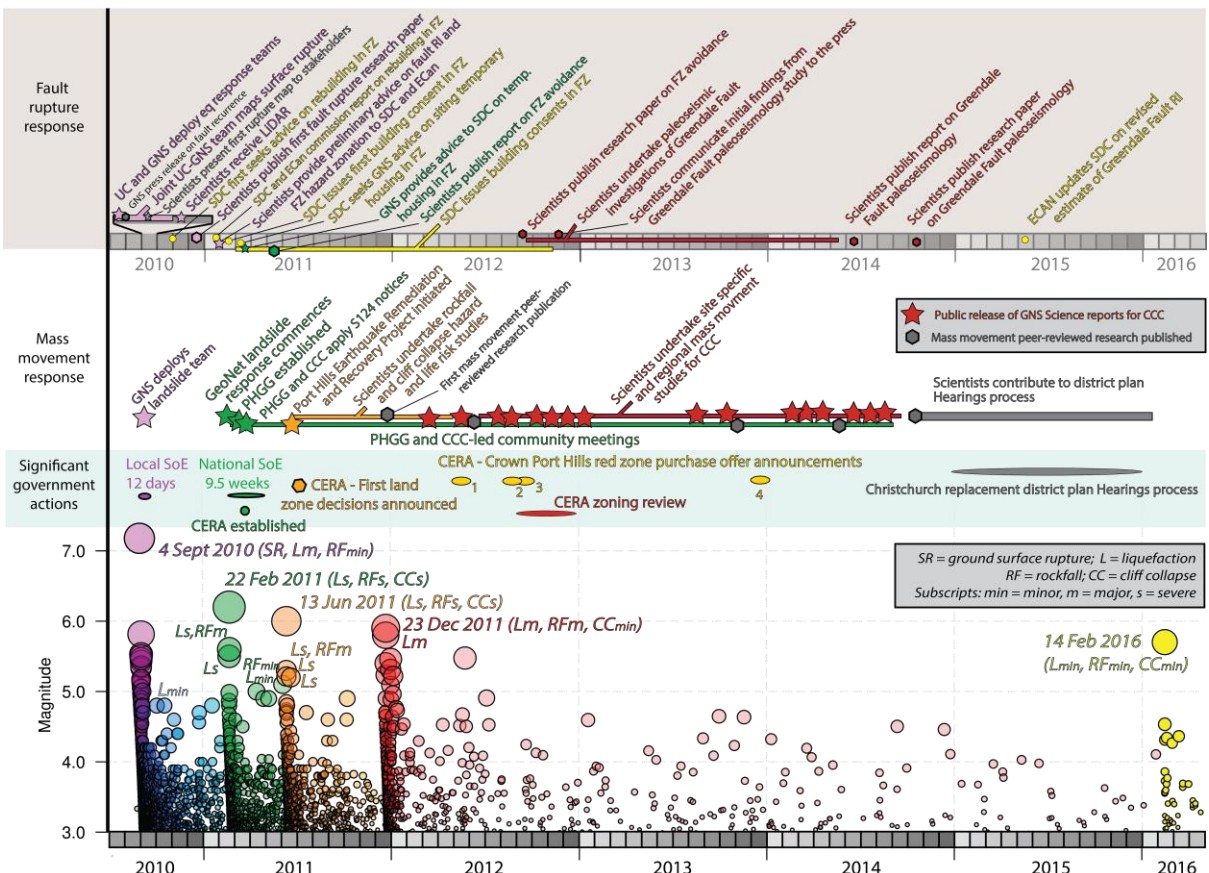

**Figure 3:** Timeline of earth science information acquisition and communication for fault rupture and mass movement plotted against major decision-making events and CES seismicity. UC = University of Canterbury; GNS = GNS Science; SDC = Selwyn District Council; ECan = Environment Canterbury; FZ = fault zone; PHGG = Port Hills Geotechnical Group; CCC = Christchurch City Council; CERA = Canterbury Earthquake Recovery Authority; SoE = State of Emergency; all other abbreviations shown in legend. See Tables 2 and S1 for supporting data.