# Peer review of "The utility of earth science information in post-earthquake landuse decision-making: the 2010-2011 Canterbury earthquake sequence in Aotearoa New Zealand"

_Natural Hazards and Earth System Sciences, 2020_

## Referee Comment (RC1) · Anonymous Referee #1 · 12 May 2020

GENERAL COMMENTS This paper is dealing with earth sciences information is used for post-disaster land-use planning decisions during the 2010-2011 Canterbury earthquake sequence (Christchurch, New Zealand). The scope of this paper is limited to mass movements and ground surface fault rupture because authors possess intimate knowledge of those hazards. Unfortunately, just brief comparisons are made for liquefaction.

Manuscript is well structured and clear. However, there are a lot of institutions involved in the research (New Zealand Cabinet, CERA, Christchurch City Council, MfE, MBIE,

[Figure]

ECQ, ECan, SDC) so a new section to explain interactions (hierarchy, competences, and so on) among them would be greatly appreciated. I would reduce the number of acronyms used (for readers not familiarised with them), especially those not used more than twice.

I would clarify the compulsory regulations, if any, to enforce geotechnical reports.

A map showing differences in land uses after CES would be appreciated to check the real impact of earth sciences information

SPECIFIC COMMENTS

Maybe answers to some of the next questions could be useful to improve the paper in order to clarify it.

Line 210 Were domestic dwellings damaged by earthquakes after 16/2/2011 (Table S1)? Line 230 Do you mean the revised Selwyn District Plan? Line 239 I suppose that buffer zone is 20 m. according to Kerr et al, 2003. Is that buffer considered in fig. 1C? Lines 285 Are Kerr et al, 2003 and Building Act the pre-disaster geotechnical guidelines? Line 305 Which is the percentage of fault avoidance zones in current district plans considering maps of past surface rupture faults? Do you have the information for revising plans? Have the results improved when compared to those published by Saunders, W.S.A., Beban, J.C. and Coomer, M.A. (2014). Analysis of natural hazard provisions in regional policy statements, territorial authority plans, and CDEM Group Plans. GNS Science report 2014/28. Lower Hutt: GNS Science? Line 360 Was the reduction of dwellings to be evacuated due to the 13 june 2011 Mw 6.0 earthquake damages? Line 426 Has reforestation been proposed as solution for rockfall hazard? Line 784. Are hazard maps legally binding? Line 801 Is District Plan including liquefaction and ground rupture?

Figure 1. I would have considered one figure for each hazard in order to enlarge some small figures.

Fig. 1A. I suppose that SDC corresponds to 1C. I have missed buildings (and specifically damaged buildings mentioned in Lines 163-166, Lines 243-245) in fig. 1C. Is the fault avoidance zone restricted to one type of deformation (well defined, distributed or uncertain) or to all of them? Is the buffer already considered?

Fig. 2 Too much information. Table S1 is more clear. Table S2 would be better with landscape orientation.

TECHNICAL COMMENTS

Line 120, 129, 146 and 148. Reference of Berryman 2012 is missing Line 280. ordinances instead of ordnances Lines 250, 251, 253 and 638. Hornblow et al, 2014 should be 2014a or 2014b? Lines 447 and 813. Building Act is not included in references Lines 490 and 682. Reports are not included in references Line 518. 5.2.3 should be 5.2.2 Line 616. Reference of Drabek 2007 is missing Line 669. "Replacement Christchurch District Plan (RCDP)" instead of "Replacement Christchurch District Plan" Line 783. Reference of Gerstenberger et al, 2104 (probably 2014) is missing Line 814 Local Government Act 2002 should be included in references Lines 821 to 825. Sentence is repeated (lines 770 to 774). Please, rewrite it. Line 845. Correct "and liquefaction hazards are the be applied"

Author contribution: W.S. contribution is missing

References: Line 901 Cubrinonovski et al, 2010 is not in alphabetical order Line 955 Hornblow et al, 2014 should be 2014a? Line 1022 Massey et al, 2011 are not cited Line 1044 Mileti, 1999 is not cited Line 1047 Neth, 2016 is not cited Line 1064 2010a instead of 2010? Line 1097 Saunders and Beban, 2014 are not cited Line 1107 Stahl et al., 2013 should be Stahl et al, 2014
* * *

---

## Referee Comment (RC2) · Anonymous Referee #2 · 13 Jul 2020

This paper is written primarily by geologists and provide recommendations to the land-use planning communities. In essence, the authors argue for more pre-planning ahead of all sorts of disasters. The authors focus on mass-movements, though many other hazards were present during the CES (i.e., liquefaction) (unless I missed something, it's unclear to me why so much emphasis was placed on mass movement instead of liquefaction).

The authors present an exhaustive account of what happened from both a geology and policy point of view during and following the CES. However, I fear that the authors have

not documented much data in the way of showing how earth science observations actually influence policy. The authors lay out numerous events in their jam-packed Figure 2, yet provide no real metrics on how valuable Earth Science information was to these decisions.

I would recommend the authors create some sort of "influence metric" that is used to figure out how useful/used ES info was at the time of decision making. No doubt, this is all included in the text, but needs to be summarized somehow and quantified. Of course, the authors point out something that really everyone knows/is common knowledge: proper preparation prevents poor performance. (not to say the performance of councils was poor–this is just a common phrase) They have an opportunity to actually show this quantitatively. More attention (perhaps another figure) should be paid to a decision made based on ES data, vs one not, and compare and contrast the outcomes.

In general, I found the manuscript a bit sprawling and challenging to retain, particularly because of the lack of figures in the text (why not include the color coded table in the Supplement, table S1, in the main text? This was far more helpful to me than Figure 2). Additionally, I found the language used throughout the manuscript quite grandiose and emphatic–word choice and tone could be softened and less polarizing.

---

## Author Comment (AC1) · 9 Aug 2020

Author response to comments: Anonymous Referee #1

We thank this reviewer for taking the time and making the effort to provide a thorough review of our manuscript. We separate their comments into RC1-1 to RC1-10 below, and respond carefully to each comment.

RC1-1: This paper is dealing with earth sciences information is used for post-disaster land-use planning decisions during the 2010-2011 Canterbury earthquake sequence

[Figure]

(Christchurch, New Zealand). The scope of this paper is limited to mass movements and ground surface fault rupture because authors possess intimate knowledge of those hazards. Unfortunately, just brief comparisons are made for liquefaction.

Author response 1: The utility of liquefaction science and engineering inputs into decision-making has been extensively analysed in our prior work (Quigley et al, 2019 – references 1,2 below) and we do not seek to duplicate that in this paper. We invited other science providers with unpublished knowledge of the liquefaction aspects to contribute to this paper and they declined. As such, the work of Quigley et al. (2019) represents the current authoritative account of liquefaction, and our choice to focus on lesser understood aspects (to-date) in this work is deliberate. Note that we do compare our study findings with those of Quigley et al. (2019 – refs 1,2) and these references are cited at several places in this manuscript. We have added a sentence to the Introduction that explicitly states why liquefaction is not the primary focus of this paper, and directs readers to Quigley et al. 2019 -1,2. REFERENCES: 1. Quigley, M.C., Bennetts, L.B., Durance, P., Kuhnert, P.M., Lindsay, M.D., Pembleton, K.G., Roberts, M.E., White, C.J., (2019) The provision and utility of earth science to decision-makers: synthesis and key findings, Environment Systems and Decisions, doi: https://doi.org/10.1007/s10669-019-09737-z 2. Quigley, M.C., Bennetts, L.B., Durance, P., Kuhnert, P.M., Lindsay, M.D., Pembleton, K.G., Roberts, M.E., White, C.J., (2019) The Provision and Utility of Science and Uncertainty to Decision-Makers: Earth Science Case Studies, Environment Systems and Decisions, doi: https://doi.org/10.1007/s10669-019-09728-0

RC1-2: Manuscript is well structured and clear. However, there are a lot of institutions involved in the research (New Zealand Cabinet, CERA, Christchurch City Council, MfE, MBIE, ECQ, ECan, SDC) so a new section to explain interactions (hierarchy, competences, and so on) among them would be greatly appreciated. I would reduce the number of acronyms used (for readers not familiarised with them), especially those not used more than twice.

Author response 2: We have included a new section (3) on Governance, which includes a table (Table 1) with the various responsibilities of all agencies described in the manuscript. The roles of these agencies are also described in:

Berryman, K. (2012). "Geoscience as a component of response and recovery from the Canterbury earthquake sequence of 2010–2011." New Zealand J. Geol. Geophys., 55(3), 313–319.

Beaven, S., Wilson, T., Johnston, L., Johnston, D., & Smith, R. (2017). Role of boundary organization after a disaster: New Zealand's natural hazards research platform and the 2010–2011 Canterbury Earthquake Sequence. Natural Hazards Review, 18(2), 05016003.

. . .and we think that interested readers can consult these papers (cited in our paper) for further information. We have further reduced acronym usage where appropriate, particularly for single usage, but have ultimately retained some acronyms in instances where word and page length would be compromised through further reductions.

RC1-3: I would clarify the compulsory regulations, if any, to enforce geotechnical reports.

Author response 3: The active fault guidelines (Kerr et al) are non-regulatory; they provide a voluntary framework for managing active faults. Geotechnical reports typically inform decision making within a planning framework, with the 'enforcement' of their content being through conditions of resource consent (if resource consent is required).

RC1-4: A map showing differences in land uses after CES would be appreciated to check the real impact of earth sciences information.

Author response 4: This is already shown in Figures 1B and 1C. Figure 1B shows the location of 'red zoned' areas that were formerly residential areas and are vacated. This is shown for both mass movement (yellow) and liquefaction (red) areas. Figure 1C shows the location of the fault avoidance zone, which was newly mapped following

the Darfield earthquake; however no land-use changes were enacted for this area, as clearly described in the text. We turn the reviewer to this figure for the information they request.

RC1-5: Specific comments (a) to (j) and Author responses:

a) Line 210 Were domestic dwellings damaged by earthquakes after 16/2/2011 (Table S1)?

Author response 5: We infer this reviewer's question to ask, 'were domestic dwellings damaged by earthquakes after the 22 February 2011 Christchurch earthquake? We see no other clear alternative to this question as stated. If our rephrasing of their question is correct, the answer varies depending on which hazard is being discussed. The text on line 201 refers to damage from fault surface rupture. Only the 4/09/10 earthquake was associated to surface rupture, and so the answer to the question is NO if referring to fault rupture. If the reviewer means to query whether domestic dwellings were damaged by earthquakes post 22 February 2011 from other hazards (rockfall, liquefaction) the answer is yes: e.g., domestic dwellings were also damaged by mass movements in the 16 April, 13 June and 23 Dec 2011 earthquake quakes. The 13 June was the most damaging. As a result of this EQ multiple dwellings were again hit by boulders, many dwellings were also damaged again by liquefaction-induced ground deformation and also shaking damage.

b) Line 230 Do you mean the revised Selwyn District Plan?

Author response: No, we refer here to the current Selwyn District Plan. The Plan is currently being reviewed (its normal 10-yearly review) and is likely to be notified to the public for submissions by late 2020. The reviewed Plan proposes restrictions on new Building Importance Category 4 and 5 buildings within the fault avoidance zone. The text has been updated to clarify this.

c) Line 239 I suppose that buffer zone is 20 m. according to Kerr et al, 2003. Is that

buffer considered in fig. 1C?

Author response: The buffer zone contains the deformation zone and the 20 m set-back. The polygon surrounding the lines on the figure (e.g., the fault avoidance zone) contains both the deformation zone and the 20 m setback. To avoid confusion, we have replaced the word 'buffer' (which we think the reviewer associated only with the 20 m setback) and use the more precise nomenclature of "fault avoidance zone". We have made changes to the text to reflect this.

d) Lines 285 Are Kerr et al, 2003 and Building Act the pre-disaster geotechnical guide-lines?

Author response: The Kerr et al, 2003, is the pre-disaster land use planning guideline for managing surface fault rupture hazard. This reference has been added to the text.

e) Line 305 Which is the percentage of fault avoidance zones in current district plans considering maps of past surface rupture faults?

Author response: Unfortunately this is beyond the scope of this paper. Many councils in New Zealand are in the process of incorporating fault information and provisions into their district plans through their 10-yearly Plan review process, but we have not done a full analysis of this; to do so would be a huge endeavour and we choose to retain the primary focus of this manuscript on to the Canterbury earthquake sequence. Our quick analysis of some District Plans indicates n=8 Operative District Plans with fault avoidance zones, n=3 Operative District Plans with active fault traces, n=2 Draft plans with fault avoidance zones. However, you can't see all the Draft Plans and we cannot check them all.

f) Do you have the information for revising plans? Have the results improved when compared to those published by Saunders, W.S.A., Beban, J.C. and Coomer, M.A. (2014). Analysis of natural hazard provisions in regional policy statements, territorial authority plans, and CDEM Group Plans. GNS Science report 2014/28. Lower Hutt:

GNS Science?

Author response: Please see the above comment. While an analysis of this type was outside the scope of this paper, as a result of the CES and planning reviews, liquefaction and slope instability are now incorporated in a far more comprehensive way than in previous plans. This paper describes how the science was incorporate into the plans, which provides a more detailed planning response for existing and future activities in these zoned locations.

g) Line 360 Was the reduction of dwellings to be evacuated due to the 13 june 2011 Mw 6.0 earthquake damages?

Author response: After the 22 February 2011, earthquake the Port Hills Geotechnical Group (PHGG) of consultants undertook a review of all 560 Red Placards on residential properties and recommended that 104 of these should not be reinstated because the life-safety risk was judged to no longer remain or be tolerable, for occupants of these dwellings, e.g., the dwelling had been demolished, or the hazard removed or the hazard and associated risk was reassessed as being low.

h) Line 426 Has reforestation been proposed as solution for rockfall hazard?

Author response: Reforestation was proposed in some areas as a solution for rockfall hazard and existing forests were considered in the risk analyses. In some areas of the Port Hills, vegetation – mainly mature trees planted close together in rows to form shelterbelts (sometime in the 1970s) – stopped boulders. Dense native forest in Lyttelton was effective at reducing the runout distances of rockfalls. However, for planning and regulatory purposes, it was decided (by CCC on advice from PHGG and GNS Science) that the inclusion of such local factors in risk assessment were problematic as in many cases the vegetation providing the mitigating effect (or land on which the forest needed to be planted) was not owned by the property it would be protecting, and neither these land owners nor the council had control over vegetation (existing or new) on private property. There was also the significant doubt locally and internationally about how effective vegetation is at stopping rockfall. There was also the issue of vegetation being susceptible to fire and storms, and ephemeral in the long-term which would render it ineffective during times of renewal. Therefore, it was decided to use reforestation as a mitigation option, where possible, but only to compliment other engineered mitigation solutions.

i) Line 784. Are hazard maps legally binding?

Author response: In general, hazard maps are only legally binding if they have been incorporated, with accompanying provisions, into a District Plan or other statutory plan through a process involving the consideration of all aspects of using the information by the council, and the opportunity for the public to provide feedback. Prior to the CES there were few District Plan provisions for liquefaction, mass movement or surface fault rupture hazards in Christchurch City, Selwyn District or Waimakariri District. We have added some clarification in lines 797-798.

j) Line 801 Is District Plan including liquefaction and ground rupture?

Author response: Yes, liquefaction and surface fault rupture can both be addressed through the District Plan. Neither the Selwyn District Plan nor the Christchurch City Plan contained provisions for surface fault rupture prior to the CES (the Christchurch City Plan still does not, because there are no fault traces at the ground surface in Christchurch). The Greendale Fault was unmapped prior to the CES, so even if their had been surface rupture hazard provisions in the Selwyn District Plan prior to the CES they would not have prevented development in the surface rupture area. Neither Selwyn District or Christchurch City had specific liquefaction provisions in their District Plans prior to the CES, although Waimakariri District did. All now have, or are proposing to have, liquefaction-specific provisions in their District Plans. We have included further wording to clarify this.

RC1-6: Figure 1. I would have considered one figure for each hazard in order to enlarge some small figures.

Author response 6: We value the reviewer's opinion but the figures are provided in high resolution and can be digitally enlarged to whatever scale the reader wishes. We have also undertaken this research without a formal budget and thus do have the funding to expand our page charges by disseminating this figure into multiple figures.

RC1-7: Fig. 1A. I suppose that SDC corresponds to 1C. I have missed buildings (and specifically damaged buildings mentioned in Lines 163-166, Lines 243-245) in fig. 1C. Is the fault avoidance zone restricted to one type of deformation (well defined, distributed or uncertain) or to all of them? Is the buffer already considered?

Author response 7: Fault avoidance zones were created for well defined, distributed or uncertain types, i.e. all types. Perhaps the confusion for the reviewer was that he/she cannot see 2 polygons (one associated with the deformation zone and another one the 20 m setback). This has now been explained in the figure caption

RC1-8: Fig. 2 Too much information. Table S1 is more clear. Table S2 would be better with landscape orientation.

Author response 8: Table S1 has been revised and inserted into the main manuscript as Table 2. Table S2 has been converted to landscape orientation, as per the reviewer's request.

RC1-9: (a) Line 120, 129, 146 and 148. Reference of Berryman 2012 is missing • REFERENCE ADDED

(b) Line 280. Ordinances instead of ordnances

• CHANGE MADE

(c) Lines 250, 251, 253 and 638. Hornblow et al, 2014 should be 2014a or 2014b? • CLARIFIED THROUGHOUT (a) or (b)

(d) Lines 447 and 813. Building Act is not included in references.

• NOW INCLUDED

(e) Lines 490 and 682. Reports are not included in references

• Included missing one; one was under Massey et al, so have changed citation

(f) Line 518. 5.2.3 should be 5.2.2 • REVISED

(g) Line 616. Reference of Drabek 2007 is missing • DELETED REFERENCE FROM TEXT

(h) Line 669. "Replacement Christchurch District Plan (RCDP)" instead of "Replacement Christchurch District Plan" • CHANGED TO REDUCE ACRONYMS

(i) Line 783. Reference of Gerstenberger et al, 2104 (probably 2014) is missing • NOW INCLUDED IN REFERENCES

(j) Line 814 Local Government Act 2002 should be included in references

• NOW INCLUDED IN REFERENCES

(k) Lines 821 to 825. Sentence is repeated (lines 770 to 774). Please, rewrite it. • SENTENCE REVISED

(l) Line 845. Correct "and liquefaction hazards are the be applied" • CORRECTED TO "liquefaction are to be applied"

(m) Author contribution: W.S. contribution is missing • INCLUDED AND SPECIFIED

RC1-10: References

(a) Line 901 Cubrinonovski et al, 2010 is not in alphabetical order • FIXED (b) Line 955 Hornblow et al, 2014 should be 2014a? • YES, FIXED (c) Line 1022 Massey et al, 2011 are not cited • REVISED (d) Line 1044 Mileti, 1999 is not cited • REFERENCE REMOVED (e) Line 1047 Neth, 2016 is not cited • REFERENCE REMOVED (f) Line 1064 2010a instead of 2010? • REVISED TO 2010a (g) Line 1097 Saunders and Beban, 2014 are not cited • Reference added to manuscript - line 73 (h) Line 1107 Stahl et al., 2013 should be Stahl et al, 2014 • CHANGED TO 2014

Please also note the supplement to this comment:
https://nhess.copernicus.org/preprints/nhess-2020-83/nhess-2020-83-AC1-
supplement.pdf

---

## Author Comment (AC2) · 9 Aug 2020

Author response to comments: Anonymous Referee #2

We thank the reviewer for making the time and effort to thoroughly review our work. We separate their review into sections RC2-1 to RC2-7 below and respond to each comment. Please note that we include figures from another paper in our response and that these figures can be viewed in the Supplement PDF to this response.

RC2-1: This paper is written primarily by geologists and provides recommendations to

the landuse planning communities.

Author response 1: The paper is co-authored by earth and social scientists with decades of experience in land use planning for hazards and risk reduction (WS), provision of natural hazard inputs and risk analyses for engineering and land use planning decisions (RvD, PV, NL, CM, MQ), and delivery of science inputs and provision of expert advice to decision-makers, including diverse government agencies, from the perspective of government (HJ, WS). The ten recommendations we offer herein are certainly not limited to land-use planning communities and in many cases they are much more aligned to earth scientists. The discussion and recommendations are actually directed towards any scientists of any affinity that wish to understand the role of, and contribute to, land-use planning prior to, during, or following the occurrence of natural hazards.

RC2-2: In essence, the authors argue for more pre-planning ahead of all sorts of disasters.

Author response 2: While we appreciate the reviewer's attempt to distil our research into a simple generic statement, this misrepresents what our work actually does. What is 'pre-planning' if the specifics of what this actually entails, diverse approaches and needs are not described, and recommendations not supported by evidence? And does the simplistic synthesis offered by the reviewer adequately encompass the ten recommendations we offer in this manuscript? In this study, we undertake a detailed analysis of how specific earth science inputs did, and did not, inform land-use decision making, including how and why they did/did not. We thus provide an evidence-base for the earth science community (including for a hierarchy in which types of earth science inputs were more used than others) that enables us to make many recommendations targeted at specific communities, for example the importance of obtaining paleoseismic data prior to or immediately following a hazard occurrence could enhance its potential utility in decision-making; not all scientists in this community will appreciate the balance of how to best meet the expedient needs of decision-makers in this regard. Pre-event

recovery planning should be undertaken with the knowledge that it's not the planning outcome per se that is important pre-event, but the process undertaken that builds relationships and understanding prior to an event.

RC2-3: The authors focus on mass-movements, though many other hazards were present during the CES (i.e., liquefaction) (unless I missed something, it's unclear to me why so much emphasis was placed on mass movement instead of liquefaction).

Author response 3: The paper clearly focuses on two significant hazards experienced in the CES; mass movements and ground surface fault rupture. It is unclear how the ground surface rupture component could have been missed; it features prominently including in a separate section (4) and in both figures of the manuscript and is of almost equal proportion to the mass movement component. It is true that liquefaction was a major hazard of the CES and required significant land-use decision-making. However, the utility of liquefaction science and engineering inputs into decision-making has been extensively analysed in our prior work (Quigley et al, 2019 – references 1,2 below). Further, we invited contributions from other science providers with inside knowledge of the liquefaction aspects to contribute to this paper and they declined. As such, the work of Quigley et al. (2019) represents the current authoritative account of liquefaction, and our choice to focus on lesser understood aspects (to-date) in this work is deliberate. Quigley et al. (2020 – refs 1,2) is clearly referenced at several places in this manuscript.

REFERENCES: 1. Quigley, M.C., Bennetts, L.B., Durance, P., Kuhnert, P.M., Lindsay, M.D., Pembleton, K.G., Roberts, M.E., White, C.J., (2019) The provision and utility of earth science to decision-makers: synthesis and key findings, Environment Systems and Decisions, doi: https://doi.org/10.1007/s10669-019-09737-z 2. Quigley, M.C., Bennetts, L.B., Durance, P., Kuhnert, P.M., Lindsay, M.D., Pembleton, K.G., Roberts, M.E., White, C.J., (2019) The Provision and Utility of Science and Uncertainty to Decision-Makers: Earth Science Case Studies, Environment Systems and Decisions, doi: https://doi.org/10.1007/s10669-019-09728-0

[Figure]

RC2-4: The authors present an exhaustive account of what happened from both a geology and policy point of view during and following the CES. However, I fear that the authors have not documented much data in the way of showing how earth science observations actually influence policy. The authors lay out numerous events in their jam-packed Figure 2, yet provide no real metrics on how valuable Earth Science information was to these decisions. I would recommend the authors create some sort of "influence metric" that is used to figure out how useful/used ES info was at the time of decision making. No doubt, this is all included in the text, but needs to be summarized somehow and quantified.

Author response 4: The authors greatly appreciate this perspective and thank the reviewer for communicating it. The request for "some sort of "influence metric" that is used to figure out how useful/used ES info was at the time of decision making" is reasonable, and the reviewer is also correct to state that "...this is all included in the text.."; indeed we have carefully considered how best to communicate our experiences and have opted for the narrative style and summary figure presented herein. Using our detailed accounts of mass movements and fault rupture hazards and decision-making, we describe in detail how different science inputs did and did not contribute to land use decision-making, and why / why not; our approach is of a highly qualitative nature. The different hazards are qualitatively compared, but more to describe the diversity of challenges encountered and how they were addressed. We retain this approach. With respect to the quantitative approach suggested by this reviewer; this approach has been previously undertaken by Quigley et al. (2020, ref 1. from above) for CES mass movement, liquefaction, and fault rupture hazards (and several other case studies from the earth sciences). Their Figure 4 (see below) provides elicited 80% confidence intervals showing each study's self-assessment in terms of science information uptake by decision-makers as a percentage (x-axis) and scientific agreement in available science inputs as a percentage (y-axis). We see no value in duplicating this analysis, and thus retain the current structure of our paper. However, we have added a statement in our paper that further directs readers to Quigley et al. (2020) for a quantitative approach

more aligned with what the reviewer suggests.

Figure 4 from Quigley et al. LINK: http://www.drquigs.com/wp-content/uploads/2019/07/Quigley2019_Article_TheProvisionAndUtilityOfEarthS.pdf

RC2-5: Of course, the authors point out something that really everyone knows/is common knowledge: proper preparation prevents poor performance. (not to say the performance of councils was poor–this is just a common phrase) They have an opportunity to actually show this quantitatively. More attention (perhaps another figure) should be paid to a decision made based on ES data, vs one not, and compare and contrast the outcomes.

Author response 5: With due respect, it is unclear how the reviewer derives this conclusion from the paper we present. Nowhere is it stated in our manuscript that "proper preparation prevents poor performance" and this statement grossly simplifies (and misrepresents) that ten recommendations provided in this paper. Indeed, one conclusion made in the paper is that more informed proper preparation (e.g., pre-disaster guidelines and collaborative networks) by earth science information providers can enhance the efficiency with which science inputs can be provided to decision-makers that require expediency, but this does not 'prevent' poor performance. Further, the types of decisions that were required to be made differed dramatically; the economic and life safety parameters and risks varied significantly, the science inputs varied, the timelines varied, and the decision-makers varied. It is not straightforward to directly compare these aspects, and please note that (i) none of the decisions made in the CES were made simply "based on ES data" in isolation from other inputs, AND (ii) none of the decisions made in the CES were made without ES data. So the binary approach suggested is not appropriate. And the outcomes are compared and contrasted throughout the text, in numerous examples. And finally, some of roles of science inputs in these decisions have already been described using a decision-tree format by Quigley et al. (2020) – see Figure below (their Figure 3; see http://www.drquigs.com/wp-content/uploads/2019/07/Quigley2019_Article_TheProvisionAndUtilityOfEarthS.pdf)

We see no reason why the highly detailed qualitative approach taken in our manuscript does not constitute a highly detailed comparative study amongst these hazards that builds upon, and provides much greater detail than, the prior work of Quigley et al. (2020).

RC2-6: In general, I found the manuscript a bit sprawling and challenging to retain, particularly because of the lack of figures in the text (why not include the color coded table in the Supplement, table S1, in the main text? This was far more helpful to me than Figure 2). Author response 6: We have opted for two main figures that synthesize our research, rather than a series of smaller figures, for two reasons: (i) This format allows all of the CES events described herein to be visually compared with each other and referenced to the same time-line within the same figure. We appreciate this figure is rich with information, but we also appreciate that disseminating this information amongst multiple figures requires constant flipping between these figures to enable comparison, which is also sub-optimal. We thus wish to retain this figure in this format. (ii) The NHESS page charges amplify significantly if we deconstruct two figures into many more. We do not wish to amplify this expense.

We appreciate the referee's feedback on supplement Table S1 and we have now included it the main text.

RC2-7: Additionally, I found the language used throughout the manuscript quite grandiose and emphatic–word choice and tone could be softened and less polarizing.

Author response 7: With due respect, this critique has little value without provision of specific examples of what the reviewer considers to be 'grandiose and emphatic' word choices, and which aspects of our narrative could benefit from softening to become less polarizing. However, given this generic comment, we have carefully reviewed the manuscript from this perspective and made 10 minor changes (word replacements).

Please also note the supplement to this comment:

https://nhess.copernicus.org/preprints/nhess-2020-83/nhess-2020-83-AC2-supplement.pdf

---

## Author Response (AR3)

We thank the reviewer for their professional review of our manuscript.

In response to the reviewer's request for provision of a statement on the (NZ) local legal framework and how that impacts on seismic hazards assessments, we add the following text to the manuscript in the Introduction:

"In New Zealand, many of the major regulatory documents and guidelines for addressing aspects of seismic hazard, such as earthquake loading standards (NZS 2004) and dam safety guidelines (NZSOLD 2015), are formulated around probabilistic seismic hazard assessment (PSHA). The prevalent use of PSHA in New Zealand has influenced the formulation of other earthquake hazard related documents such as the Ministry for the

Environment's "Active Fault Guidelines" (Kerr et al., 2003) which uses surface rupture recurrence interval (as a proxy for annual exceedance probability). For landslide hazards there are no New Zealand-specific regulatory documents to guide the practitioner with respect to the appropriate method to analyse the hazard or what risk metrics to use and/or what risk thresholds a regulator should consider as intolerable/tolerable. Within New Zealand, Saunders and Glassey (2007) provided guidelines on landslide hazard and risk for planners (with use applications relevant to geological and geotechnical practitioners, and developers) for resource consent applications and planning documents at regional and district levels. The Australian Geomechanics Society (AGS) 2007 landslide risk management guidelines are widely used within New Zealand but not officially adopted by any agency. The AGS started to develop guidelines for landslide risk management in the 1990's and subsequently updated in 2007 (AGS, 2007). At the same time and in conjunction with the AGS, the Joint Technical Committee-

1 (JTC-1) comprising members from the International Society for Soil Mechanics and Geotechnical Engineering (ISSMGE), the International Society for Rock Mechanics (ISRM), and the International Association of Engineering Geologists (IAEG) developed guidelines on landslide susceptibility, hazard and risk zoning for land use planning (Fell et al., 2008a), to set recommended international best practice. To a large extent, this is what has been adopted in the mass movement hazards work covered in this investigation."

We also note, but do not comment further in the manuscript, the following:

Different agencies have differing needs and wants, and thus the area-wide mass movement assessments were based on quantified risk analysis and assessment methods as set out in AGS (2007) and Fell et al., (2008). These guidelines stipulate that for landslide risk analysis the frequency of landslides affecting a site, should be assessed from multiple data sources, in order to establish the frequency (often annualised) of a given magnitude of landslide occurring, for the different types of landslide (e.g., rockfall, debris avalanche etc) that could affect a given site. These magnitude frequency relationships should cover the full range of landslide magnitudes and their likely triggering events including earthquakes and non-earthquake triggers such as rain. As a result, we do not use a single annualised frequency/annual exceedance probability of e.g., a single earthquake event occurring and the consequences associated with that single event. Instead, the risk analysis method comprised the following main steps:

1)     Consider the possible range of triggering events in terms of a set (bands) of earthquake triggers and a set of non-earthquake triggers (e.g. rain, time, etc.).

2)     For Earthquake Triggers:

a.      Choose a small set of representative events (earthquake bands) spanning the range of event severity (using the peak ground acceleration), from the lowest to the highest – we used four bands from 0.2 to 0.4g, >0.4 to 1g, >1 to 2g and >2g; b.      For each representative event (band), estimate the number of landslides of a given volume class produced in that event.

c.      Estimate the annual frequency of the representative earthquake event (per band) occurring, from the Probabilistic Seismic Hazard Model for New Zealand.

3)      For Non-Earthquake Triggers:

a.      Choose a small set of landslide volumes that span the range that could occur and, b.      for each representative landslide volume class, calculate the annual frequency of the landslide occurring from historical and geomorphological records.

These frequencies were annualised, and used in the risk analysis, along with: 1) estimates of the probability of debris reaching or passing a given location and the probability of a person at that location being in the path of debris; 2) the probability that a person is present in the given location as the debris moves through it; and 3) the probability that a person is killed if present and in the path of debris. The risk from all events was then summed to estimate the combined risk from earthquake and non-earthquake landslides of a given type. Christchurch City Council (CCC) and the Canterbury Earthquake Recovery Authority (CERA) both adopted the results from the risk analysis method, but the inputs used were "tweaked" based on the requirements of each agency. These are detailed below.

We have also amended the reference list as indicated.

Sincerely,

Mark Quigley and co-authors

[revised manuscript text omitted]